# History of the Potsdam, Seddin and Niemegk Geomagnetic Observatories – Part 3: Niemegk

**Hans-Joachim Linthe[1]**

[1] {Helmholtz Centre Potsdam GFZ German Research Centre for Geosciences}

Correspondence to: Hans-Joachim Linthe (linthe@gfz-potsdam.de)

**Abstract**

The measurement series of the 3 geomagnetic observatories Potsdam, Seddin and Niemegk span over more than 130 years, starting in 1890. It is one of the longest, almost uninterrupted series of recordings of the Earth's magnetic field. Data users frequently emphasize the high quality of the data and its significance for geomagnetic base research. The very well-known and outstanding scientists for geomagnetism, Max Eschenhagen, Adolf Schmidt, Julius Bartels, Gerhard Fanselau and Horst Wiese directed among others, in historical sequence, the three observatories.

This paper describes the history of the Niemegk Adolf Schmidt Observatory, which was started in 1932 and is currently still in operation.

## 1 Introduction

The Potsdam Prussian Meteorological-Magnetic Observatory operated two magnetic observatories: Potsdam on the Telegrafenberg near by the town of Potsdam, which was in operation 1890-1928, and Seddin near by the village of Seddin, which worked 1907-1931. Anthropogenic noise from DC-powered railway traction vehicles, which were operated in small distances, caused the termination of both the magnetic observatories (Best, et al., 1991 and 1992; Linthe, 2023a and 2023b). It was necessary to find a suitable location for a new observatory, which allowed the undisturbed long-term operation.

Adolf Schmidt was successful in doing this job. The mayor of the town of Niemegk offered excellent conditions for the new observatory. The construction of the observatory buildings

took place under extreme time pressure – Potsdam and Seddin were already disturbed by the DC-powered railway traction vehicles.

The next problems occurred by upwelling ground water in the variation house, which caused comprehensive construction repairs. The end of World War II in 1945 caused building and instrument damages and – mostly serious – the lost of instruments. The observatory personnel restarted the observations under extremely difficult conditions, but they could not prevent a data gap of 9 months.

The following decades were years of precise observations, successful scientific research, fruitful international collaboration and excellent instrument construction. The German reunification in 1990 implicated dramatic changes for the observatory. The number of employees decreased extremely, but the conditions of the operation of the observatory improved strongly. Modern instruments were purchased and the access on up to date IT hard and software rendered possible to make the observatory completely digital.

The International Kp Index Service was taken over from Göttingen University in 1997. Actually it was the return home – the K index was invented on the base of Niemegk recordings in 1939. Even scientific conferences were performed at the observatory. Two employees of the observatory were awarded by the log-service IAGA medal.

In the beginning of the data series Potsdam-Seddin-Niemegk the observatories belonged 1890 - 1932 to the Magnetic-Meteorological Observatory Potsdam. From 1933 – 1991 followed a period of several affiliation chances. From 1992 onward a period of constant affiliation to the GeoForschungsZentrum Potsdam – now termed Helmholtz Centre Potsdam GFZ German Research Centre for Geosciences - followed.

The most important fact for the observatory are the unchanged magnetically clean observing conditions from the beginning until today, therefore, Adolf Schmidt selected an excellent place for the observatory. Furthermore, the administration of the town of Niemegk accepted over the complete operation time – almost 100 years – the agreement stating the conditions of an undisturbed operation.

The paper contains the following sections:

1. Introduction
2. Niemegk Adolf Schmidt Geomagnetic Observatory
   2.1 Niemegk observatory buildings

The paper is completed by seven appendices, competing interests and acknowledgements:

Appendix I – Brief description of observatory instruments

Appendix II – Further observatory equipment

Appendix III – Activities of the observatory, exceeding its ordinary purpose

Appendix IV - Selection of significant meetings and conferences related to the observatory

Appendix V – Collaboration with international observatories

Appendix VII - Prominent scientific results and instrumental achievements connected with the observatories Potsdam – Seddin – Niemegk

Competing **i**nterests

Acknowledgements

## 2   Niemegk Adolf Schmidt Geomagnetic Observatory

The famous Adolf Schmidt (1860-1944), director of the magnetic observatories 1902-1928 tackled the task to find a suitable place for a new observatory. At first he considered locations near the villages of Raben and Rädigke in the Hoher Fläming hills (Bock, 1950). But the power supply and building heating was a problem at that time at those locations. But due to a favourable circumstance Adolf Schmidt found a suitable location. Paul Temming (1884-1953), mayor of the small town of Niemegk 1917-1937, got by chance knowledge of Adolf Schmidt's intension to establish a magnetic observatory in the region of Hoher Fläming. He expected development impulses for Niemegk, having such a facility in the town. He made aware Adolf Schmidt on his idea. Schmidt brought forward his expectations on the location and the conditions of the undisturbed operation of the observatory. The negotiations of both the authorities came to the successful result to establish the new observatory in a distance of 1,000 m from the town boundary at the edge of a forest (Nippoldt, 1929). On Adolf Schmidt's request the town of Niemegk and the observatory agreed in a contract to ensure the

undisturbed operation of the observations. The town of Niemegk committed to abstain from DC facilities and to accept a protection circle of 500 m radius around the observatory, in which any constructions should be allowed only in case of the permission by the observatory. Fig. 1 shows this contract.

The town of Niemegk bought the properties from private owners and sold it for the half of its price to the observatory. Furthermore the town constructed a gas and water pipeline and the electrical power supply connection to the observatory, bearing one third of the expenses (Bock, 1950).

It was intended to start and finish the construction of the new observatory in 1927 to enable the parallel operation of Seddin and Niemegk over a suitable time period. The Deutsche Reichsbahn-Gesellschaft, owner of the Berlin suburban railway, announced to start the electrical operation of the trains at the beginning of 1929. In fact it was started already on 11 June 1928. But the construction of the Niemegk observatory started only in 1929 and became functioning only in the course of 1931. The parallel operation of Seddin and Niemegk was only possible during the time period of almost one year, but the Seddin observations were already disturbed by the leakage currents of the electrical trains (Linthe, 2023b).

Adolf Schmidt continued to accompany the activities even after going retired on 1 October 1929. He actively influenced the construction plans of the buildings and their locations. The Deutsche Reichsbahn-Gesellschaft, the operator of the DC powered Berlin suburban railway, as the originator of the movement of the observatory from Potsdam and Seddin to Niemegk contributed a significant fee to the costs of the new observatory on the base of a contract – as a result of Schmidt's successful negotiations (Nippoldt, 1930; Best, 1997).

The construction activities of the observatory started on 3 May 1929 (Nippoldt, 1931b). The Prussian Minister for Science, Art and Education granted on 1 April 1930 the new observatory the name "Adolf-Schmidt-Observatorium für Erdmagnetismus". Fig. 2 shows the document. On 23 July 1930, Adolf Schmidt's 70[th] birthday, the observatory was officially opened (Nippoldt, 1931a). Fig. 3 shows the first page of the observatory guest book with Adolf Schmidt's inscription. He wrote his motto "To be always excellent and to distinguish from others" in Greek and old-style German. Forty six invited persons participated in the opening ceremony. At Fig. 4 the inscriptions of all of them in the guest book are depicted.

After the complete finishing of the building construction in 1931 the instruments were installed and adjusted by Richard Bock (1899-1961). The operation of Seddin observatory

was continued until 9 May 1932. On 1 January 1932 the observations started officially at the Niemegk Adolf Schmidt Observatory (Bock, 1950).

**2.1 Niemegk Observatory Buildings**

Fig. 5 shows the ground plan of the new observatory compound of a size of 3.5 ha (Bock, 1939). The solid clinker-constructed apartment and service building (later called main building) contains 2 apartments, some offices, 2 guest rooms, the necessary power supply equipment, a workshop and a central heating system. In the as well as solid clinker-constructed storage house were a garage, a laundry and storage rooms. Fig. 6 shows a photo of both the buildings.

The house for absolute measurements (later called absolute house) and the variation house are of wooden construction on a concrete foundation. The sand, cement and further additives, used for the concrete, were carefully investigated on being free of any magnetic influence on the observations. Any fitting assemblies are made from brass, the nails are copper ones. The pillars for the instruments are as well as concrete ones, resting in groups together (variation house) resp. all together (absolute house) on a concrete foundation of 50 cm thickness. An outdoor pillar is located eastward of the absolute house.

The absolute house (see a photo at Fig. 7 and the ground plan at Fig. 9) consists of a crawl space below the Earth's surface and the measurement room at the ground floor with another crawl loft for thermal insulation. Sixteen pillars are available for the vibration-free placement of instruments (14 in the main room and another 2 in a special annex chamber at the west wall). The church tower (viewable at Fig. 12) and the water tower of the town of Niemegk in a distance of about 1,000 m, visible from some of the pillars, serve as azimuth marks. Before the wooden construction of the house was started, astronomical measurements to determine the azimuth values of a number of inside pillars and the outside one with respect to the azimuth marks took place (Nippoldt, 1930).

In 1957 inner walls with thermal insulation and a thermoelectric controlled heating was added in the absolute house to ensure a constant temperature for the absolute measurements (Wiese, 1960a). All this was removed in 2003 and 2004 in connection with the basic renovation of the building. Since the invention of the new set of absolute measurement instruments from 1992-1996 a strict constant temperature during the absolute measurement set was not any more necessary.

The variation house (see a photo at Fig. 8 and the ground plan at Fig. 10) partly sunk-in below the Earth's surface to avoid as good as possible a thermal influence on the instruments. Around the instrument rooms an insulation corridor for further thermal protection takes course. Three rooms, called following their geographic directions – south, north-east and north-west – are intended to be used for the instruments. A service corridor in the centre allows the access on the recording equipment of both the northern rooms and to enter the south room. The wooden loft with roof over all the concrete basement is further intended to avoid thermal influence on the instruments. A crawl space below the wooden floor of the instrument rooms and the service corridor allows the placement of cables.

Adolf Schmidt intended a separate heating system on the base of gas for the variation house and tile stoves for the absolute house. But finally a central heating system for the heating of both magnetometer buildings was constructed. A separate solid clinker-constructed heating house (Fig. 11 shows a photo of the building) containing the coal burning boiler, the pumps and the coal storage in a suitable distance to the observation buildings was in use during the cold seasons. The connection pipes and the radiators in the observation houses were made from copper. Only the insulating corridor of the variation house is heated by means of the warm water heating system. The instrument rooms are heated by means of thermoelectric controlled electric radiators. The annual temperature variation in the rooms is less than 0.5 degrees centigrade.

The two Seddin observatory buildings were re-erected on prepared magnet-free concrete foundations in 1932 (Nippoldt, 1934). The former Seddin variation house was intended to be used as a laboratory, which is still its function nowadays. It was also connected to the central heating system operated from the heating house. Its basement has a crawl space for cable placement. The former Seddin absolute hut was called first "observation hut". Later it was renamed into "small hut". A photo of its present situation shows Fig. 5 in Linthe, 2023b. Fig. 12 shows a photo of the observatory buildings, viewed from the apartment and service house.

In 1960 the construction of a new building as an electric laboratory was finished (Fanselau, 1962a). Fig. 13 shows a view from the attic floor of the main building on the laboratory. In 1961 two more measurement huts were constructed at the observatory compound, one of them for housing the sensor of the proton magnetometer. In 1963 an observation hut for the telluric recordings and a little solid building for housing of a small computer was added (Fanselau,

1964). During the next year a wooden building housing the measurement centre was constructed.

In 1967 the construction of a solid house, called workshop building, housing the precision mechanical workshop, 3 guest rooms, 3 offices, 2 meeting rooms, a kitchen and a photographic laboratory was finished  (Fanselau, 1968). Fig. 14 shows a photo of the building. In the same year the computer building was enlarged.

Further buildings were constructed at different times for different purposes. Finally 26 buildings existed on the observatory compound. Fig. 15 shows the ground plan of 2003. One of the 26 buildings did not any more exist at this time. Three more buildings, which were not any more in use, were removed in 2004. Fig. 16 shows the present ground plan of the observatory compound.

The buildings No. 1, 4, 5, 6, 7, 8, 9, 10 and 11 were declared historic monuments in 2004.

## 2.2   Niemegk Observatory Instruments

The before in Potsdam and in Seddin used instruments were transported step by step to Niemegk and installed at the new site. Brief descriptions of the instruments and the measurement procedures can be found in appendix I.

### 2.2.1   Absolute Measurements

The three theodolites for the measurement of declination and horizontal intensity, Wanschaff, Bamberg and Schmidt as well as the oscillation box Wanschaff have got suitable conditions in the new absolute house due to the big number of 14 pillars. The 2 Earth inductors Schulze No.1 and No. 65 as well as the appropriate galvanometers have got their places on separate pillars. Fig. 17 shows an interior view of the absolute house. The table below the figure gives the assignment of the visible instruments to the pillars.

The accuracy of the oscillation measurements was significantly improved by means of the construction of a photoelectric oscillation measurement facility in 1933 (Fanselau, 1933).

The theodolite Bamberg, the oscillation box Wanschaff, the photoelectric oscillation measurement facility and the standard magnets for the measurement of the horizontal intensity were lost in 1945 (Fanselau's preliminary remark in Richard and Wiese, 1954). New standard magnets were purchased in 1950. A new oscillation box was constructed by the observatory workshop. The lost photoelectric oscillation measurement facility was replaced

by a new established one in 1954 (Schmidt, 1956). A further Earth inductor was delivered by Mating & Wiesenberg, Potsdam (Richard and Wiese, 1954). The new positions of the instruments in the absolute house were as follows:

| Pillar No. | Instrument |
|---|---|
| 2 | Galvanometer for the earth inductor Mating & Wiesenberg |
| 4 | Galvanometer for the earth inductor Schulze No. 1 |
| 6 | Collimator (azimuth mark in case of invisible towers) |
| 7 | Oscillation box |
| 8 | Theodolit Wanschaff |
| 9 | Theodolit Schmidt |
| 11 | Earth inductor Mating & Wiesenberg |
| 13 | Earth inductor Schulze No. 1 |

From 1959 onward several self-made proton magnetometers were in use for permanent measurements of the total intensity (Schmidt, 1962; Wiese, 1962). In 1970 a Russian caesium magnetometer was taken into use on pillar No. 1 in the absolute house. Comparisons to the proton magnetometer measurements were done (Lengning et al., 1973). Later on a self-made vector proton magnetometer was installed on pillar No. 1 in the absolute house.

In 1992 two DI-flux instruments on the base of the theodolite ZEISS THEO 010B equipped with the Bartington flux-gate magnetometer MAG01H were purchased and taken into use. Fig. 18 (left) shows the DI-flux on pillar No. 2 of the absolute house with the Niemegk church tower in the background. The absolute measurements by means of these instruments were performed on pillar No. 2 and 5 in the absolute house. Measurements of the total intensity were carried out on the same pillars by means of the Overhauser proton magnetometer GEMSYS GSM19 (Best et al. 1993). The GSM19 is depicted at Fig. 18 (right).

Brief descriptions of the instruments and the measurement procedures of the modern instruments can be found in appendix I as well.

## 2.2.2 Variation Recordings

In the beginning variometers, formerly operated in Potsdam or Seddin were installed in Niemegk. The differences of Potsdam and Seddin against Niemegk are as follows: Instead of baseline interruptions (Potsdam and Seddin) vertical lines on the photographic recordings were in use as hourly time marks in Niemegk, controlled by the non-magnetic pendulum clock, which was moved from Potsdam. In Potsdam and Seddin gasoline lamps were in use

for the illumination of the photographic recording. In Niemegk from the beginning electrical lamps powered by batteries in the main building were used exclusively for the photographic recording.

The variometers of Mascart's origin, used before at the Potsdam observatory for the recording of the declination (D), horizontal (H) and vertical intensity (Z), were mounted after some modifications and alignments in the north-east room of the Niemegk variation house. An instrument for recording of the inclination (I) was added. The projected sensitivities were 2 nT mm$^{-1}$ for H and Z and 0.4 arc minutes mm$^{-1}$ for D and I. The recording equipment (made by Askania, Berlin-Friedenau), having 4 drums for the photographic paper, used 2 of the drums for 2 variometers each on 1 paper sheet of a recording speed of 2 cm hour$^{-1}$. The 2 further drums allowed the recording of the same variometers of selectable recording speeds of 6 or 24 cm hour$^{-1}$ (Bock, 1935).

Fig. 21 shows one of the first test photographic recordings of the horizontal intensity, declination and the vertical intensity of the time interval 25 March 1931 at 08:00 till 26 March 1931 at 07:20 (Greenwich local mean time) taken at the Niemegk Adolf Schmidt Geomagnetic Observatory.

The former in Seddin used variometer set including the photographic recording equipment for recording of the North (X), East (Y) component and the vertical (Z) intensity was placed in the Niemegk north-west room in the same disposition as at Seddin. Fig. 19 shows an interior view of this room. The tracks on the photographic recordings are described in Linthe (2023b) and remained unchanged.

All the variometers in the north-east and north-west room were equipped with Helmholtz coils for the galvanic scale value determination.

In the south room a declination (D), horizontal (H) and vertical intensity (Z) variometer set was operated. From 1937 onward a special recording unit (Schmidt, 1926) of a plotting speed of 4 mm minute$^{-1}$ was operated, which was already in use in Seddin (Linthe, 2023b).

From 1938 onward an instrument set of reduced sensitivity (25 nT mm$^{-1}$ for H and Z and 5 arc minutes mm$^{-1}$), a so called storm variometer, was operated in the west room of the magnetic laboratory.

Caused by the World War II damages of buildings and instruments and loss of 3 variometer sets in 1945, only a provisional operation of the north-east system was restarted in February

1946. Details are described in chapter 2.3 at page 12. Fig. 22 shows the first photographic recordings after the observation gap of the horizontal intensity, declination, the vertical intensity and the room temperature of the time interval 27 February 1946 at 10:30 till 28 February 1946 at 09:00 (Greenwich local mean time). The principle of recording the variations of different components of the Earth's magnetic field and temperatures at the same magnetogram, which is clearly visible at this figure, was in use all the time at the Niemegk observatory.

In 1948 a new set of variometers of reduced sensitivity (storm variometer) was installed (Fanselau and Wiese, 1956). In 1950 a new variometer set was installed in the south room of the variation house, recording the North (X), East component (Y) and the vertical intensity (Z), made by Mating & Wiesenberg, Potsdam. Also in 1950 a new variometer system was installed in the north-west room, recording H, D and Z, which was made in the observatory workshop.

The operation of the storm variometer in the magnetic laboratory was stopped in 1951 and replaced by a journey recording unit (Fanselau, 1951), which was operated additionally in the north-west room of the variation house. It was finally replaced by a new H, D, Z storm variometer set in the north-east room of the variation house in 1954 (Wiese, 1957).

In 1960 a new recording equipment made by Mating & Wiesenberg, Potsdam was installed for the north-west variometer system.

In 1965 a special paper-economizing three-component photographic recording using an especially constructed recording equipment was started in the magnetic laboratory (Fanselau and Grafe, 1963). It was in operation until 1970 (Lengning et al. 1971).

Digital recording vector proton magnetometers were constructed during the 1970[th] at the observatory. Their continuous operation at Niemegk observatory and in the remote station Warnkenhagen (at the Baltic Sea cost, north-west German Democratic Republic, GDR) started in 1976 (Lengning et al. 1977). In 1978 a digital recording scalar proton magnetometer was installed at the remote station Sosa in the Erzgebirge (south GDR). All the proton magnetometers recorded of a sampling rate of 1 minute on eight canal punched tapes. The operation of these instruments at the remote stations was terminated in 1991. The termination of the Niemegk vector proton magnetometers followed in 1994.

In 1993 an "automatic geomagnetic observatory" M390, made by the French company GEOMAG, consisting of the fluxgate magnetometer VM390A, the Overhauser proton magnetometer GEMSYS SM90R, the electronic unit and a METEOSAT transmitter was installed and operated continuously in the variation house. The recorded data were transmitted to INTERMAGNET and stored on a 3.5" floppy disk. The operation of this instrument was terminated in 2006.

In 1995 the 3-component fluxgate magnetometer FGE, made by the Danish Meteorological Institute Copenhagen, and the Overhauser proton magnetometer GSM19, made by GEM Systems, Richmond Hill, Canada, were taken into operation in the variation house. They were controlled by the self-made data logger MAGDALOG (Best and Linthe, 1996). Fig. 20 shows a photo of the FGE. The GSM19 is depicted at Fig. 18, right. Two more of this digital recording system of the same configuration were installed in the variation house in 2000. These instruments are the present variation recording systems of the Niemegk Adolf Schmidt Geomagnetic Observatory. Only the GSM19 were replaced by the Overhauser proton magnetometers GSM90. The vector sensors FGE are located in the south room, the scalar sensors are placed in the north-east room of the variation house. A further digital recording system was installed in 2008 in an underground container (No. 16 at Fig.16).

In 1996 a further 3-component fluxgate magnetometer, made by MAGSON, Berlin, was installed in a measurement hut. The recorded data were transmitted manually by means of a laptop (Best and Linthe, 1997). The operation of this instrument was terminated in 2005.

The photographic recordings were continued until the photographic paper was finished. Fig. 23 shows the last photographic recordings of the north (X), east (Y) component and the vertical (Z) intensity and 2 room temperatures of the time interval 27 May 2006 at 09:00 till 28 May 2006 at 9:00 (Greenwich local mean time)  taken at the Niemegk Adolf Schmidt Geomagnetic Observatory.

Brief descriptions of the classical and modern instruments for recording of the variations can be found in appendix I as well.

## 2.3   Operation of the Niemegk Observatory

On 30 May 1930 a caretaker and a technician moved into their apartments of the main building. Richard Bock (1899-1961) as the observer followed them on 1 December 1930

(Nippoldt, 1931a). These 3 employees operated the observatory on-site. All the data evaluation took place at the institute head quarter in Potsdam.

Already in 1931 ground water welled up in the variation house, which dramatically degraded the operation conditions of the instruments. The seasonal changing level of the ground water was not carefully enough considered during the planning phase. An expensive and time consuming drainage construction (finished in November 1931) drained off the ground water (Bock, 1950). Together with a ventilation system the situation for the instruments was finally improved. But the wooden beams and shelves of the floor construction were so aggrieved that in 1934 a chemical conditioning of the beams and replacement of the shelves was necessary (Bock, 1937 and 1950). These measures required to remove all the instruments from the building. The replacement recordings took place in the magnetic laboratory, which was taken into use end of 1933 after its demolition in Seddin and re-erection in Niemegk. After a more or less provisional operation 1931-1934 of the observatory due to the construction defect of the variation house finally from 1935 onward a normal situation started.

The absolute measurements were performed by means of the theodolite Wanschaff on pillar No. 9 for the declination (D) and horizontal intensity (H), supplemented by the oscillation box on pillar No. 3 and by means of the Earth inductor Schulze No. 1 on pillar No. 11 for the inclination (I). The associated galvanometer was placed on pillar No. 2.

From 1936 onward plots of typical variations as Bay disturbances, sudden storm commencements (ssc) and further characteristic trends of the geomagnetic variation field were included into the yearbooks. In 1937 plots of pulsations followed. For the publication in the yearbooks the photographic recordings were scale-transformed using a special developed pantograph, constructed by Adolf Schmidt (Luyken, 1909).

From 1937 onward the magnetic activity indices K, proposed by Julius Bartels (1899-1964), were published regularly in the yearbooks (Bartels, 1938). The index was internationally adopted on Bartels' suggestion at the Washington conference of the International Association for Terrestrial Magnetism and Electricity (IATME) in 1940.

Already in 1944 World War II influenced the Niemegk territory. Bombings and airplane shots attacked the town. The storm on the town of Niemegk by Soviet tank, artillery and infantry forces took place in April 1945 (Dalitz, 1995). The last magnetograms were taken off the recording equipment on 20 April 1945. The absolute house was heavily damaged by an artillery strike, whereby the instruments were totally contaminated. A further artillery strike

damaged the transformer house, so that the power supply was interrupted until September 1945. The most serious consequence was the instrument loss, commandeered by the victor force (Fanselau and Wiese, 1954).

Only under strenuous efforts the war damages were abolished step by step. The operation of the observatory needed to be re-established completely anew. The artillery strike of the absolute house caused a lot of shrapnel in the wooden parts, which needed to be individually and extensively discovered and removed additionally to the building repair (Fanselau and Wiese, 1956). The variometer recordings were restarted on 27 February 1946, first only provisionally. Due to the loss of the standard magnets the absolute measurements of the horizontal intensity were performed using a magnet of low quality.

On the newly purchased or constructed instrumental base a new determination of the absolute level of the Earth's magnetic field values took place 1950-1952 (Richard and Wiese, 1954). In this connection the azimuth values of the outdoor pillar, of some of the pillars in the absolute house and both of the ones in the small hut with respect to the Niemegk church and water tower and further distant village church towers were geodetically newly determined.

Up to this time only theodolite Wanschaff was in permanent use for the determination of the declination and horizontal intensity. From this time onward these measurements were performed by means of both the theodolites, Wanschaff and Schmidt. Their results were averaged.

On the base of the experimental studies of proton magnetometers, started in 1950, an equipment for the permanent measurement of the total intensity was established (Schmidt, 1962; Wiese, 1962). It was in use from 1958 onward. The results were published from 1959 onward in a special yearbook table, demonstrating the difference of the proton magnetometer measurements to the classical ones. The total intensity measurements were performed manually operated on the outdoor pillar "Waldpfeiler" ("forest pillar") in hut No. 15 at Fig. 15) as well as in the absolute house on pillars No. 15 and 16 regularly on workdays. From 1965 onward such measurements were in parallel carried out also in the absolute house on pillar No. 2. The data of the proton magnetometer measurements of all installed instruments were permanently compared. In 1962 a survey of the total intensity in the absolute house was carried out (Schmidt, 1963) to find out magnetic anomalies. Neglectable anomalies were found.

The observatory results on the base of the proton magnetometer measurements were of higher accuracy in comparison to the data achieved from inclination measurements by means of the Earth inductor. Therefore, consequently from 1966 onward the measurements of the total intensity by means of proton magnetometers were directly used for the baseline calculation. The inclination measurements by means of the Earth inductors were used only for level check and finally terminated (Grafe, 1968).

In 1968 the first magnetic recording instruments with digital output were taken into operation (Fanselau, 1969).

## 2.4 Development of the Observatory after the German Reunification in 1990

After the positive evaluation of the Niemegk Adolf Schmidt Geomagnetic Observatory by the German Council of Science and Humanities it was decided to integrate the observatory into the GeoForschungsZentrum (GFZ) Potsdam, which was founded on 1 January 1992.

The observatory started in 1931 with 3 on-site employees. This situation did not change until the time immediately after World War II. Gerhard Fanselau lost his apartment in Berlin due to bombing attacks on the city. He took a free apartment at the observatory. He first arranged the repairs of the demolished buildings and instruments and restarted the observation service. Next he promoted a comprehensive development of the observatory. He initiated the instrument development and established a scientific working group in Niemegk. He looked after the logistic base and recruited the necessary number of employees: technicians and scientists. Even after Fanselau's retirement in 1969 the number of employees increased up to 55 persons during the eighties. With the foundation of the GeoForschungsZentrum (GFZ) the employees number decreased dramatically, but more or less social compatible. The Unification Treaty determined the closing of all institutes of the Academy of Sciences of the German Democratic Republic on 31 December 1991. New positions for scientists and technicians were opened during the foundation of the GFZ in the course of the year 1991. Former employees of the observatory, who were not considered for any new observatory position went retired, changed to other enterprises, took alternative positions within the GFZ or took project positions.

From 1992 onward all the historical computers were replaced step by step with modern personal computers. The complete variation recording was transmitted into the digital mode by installing self-made data loggers MAGDALOG and glass fiber data transmission. The complete observatory data are data-base stored and secured by suitable backup systems.

From 1992 onward absolute measurements were performed regularly by means of the DI-flux (declination, inclination) and the GEMSYS GSM 19 (total intensity) in parallel with the classical ones. The results of both measurements were compared.

The Niemegk Adolf Schmidt Geomagnetic Observatory became an IMO (INTERMAGNET Geomagnetic Observatory) in 1993. In 1994 the survey of the total intensity of the pillars in the absolute house was repeated (Linthe, 1995). The new survey confirmed the results of (Schmidt, 1963).

From 1994 onward digital observatory data were published on 3.5" floppy disks besides the yearbook tables (Best and Linthe, 1995).

After the successful comparisons of the absolute measurements by means of the classical instruments and the modern ones over 4 years in 1996 the classical absolute measurements were stopped. The observatory level was based from this time onward on the DI-flux and the Overhauser proton magnetometer on pillar No. 8. This instrument change caused a jump of the observatory level in the horizontal and vertical intensity. The classical photographic recordings were continued. But the observatory data were based on the digital recordings of the 3-component fluxgate magnetometer FGE (Best and Linthe, 1997).

A new determination of the azimuth values of the outdoor pillar and several absolute house pillars was carried out in 1997 (Förster, 1998).

The Bundesamt für Seeschifffahrt und Hydrographie Hamburg (BSH) decided to terminate the operation of the Erdmagnetisches Observatorium Wingst with 1 January 2000. The GeoForschungsZentrum Potsdam (GFZ) and the BSH agreed in a contract to continue the observations in Wingst. The BSH remained responsible for the management of the compound and the buildings, while the GFZ took over the operation of the instruments and the scientific responsibility (Linthe and Schulz, 2005). Wingst Observatory was finally taken into complete responsibility of the GFZ in 2014. From 2000 onward joint yearbooks of both observatories were published. The yearbook publication was terminated with the 2003 one.

## 2.5    Affiliations, Observers and Directors resp. Heads of the Observatory

The Niemegk Adolf Schmidt Geomagnetic Observatory was affiliated to different scientific or administrative organisations. Table 2 shows the complete affiliation history. Table 3 contains the list of the scientific directors resp. heads of the observatory. The responsible observers are listed in table 4.

## Appendix I – Brief Description of Observatory Instruments

Magnetic observatory data can be achieved only by combining continuous recordings of the temporal variations of the Earth's magnetic field with periodical absolute measurements. The absolute measurements are necessary to calibrate the variation instruments (variometers). The classical instruments, which were in use in Niemegk until the 1990s, were constructed mainly after the principle of a suspended magnet.

Variometers for recording of horizontal elements, for instance the North (X), or East component (Y), or the horizontal intensity (H), or declination (D) consist of a bar magnet suspended by means of a vertical thread. The magnet can only move in one direction, corresponding to the element to be recorded. The instrument for the vertical component (Z) needs a magnet able to move vertically. Practically this is achieved by mounting the magnet on a balance. Both the horizontal and vertical variometers have a mirror at their face side.

During the beginning of magnetic observatories the position of the magnet was manually read by means of a telescope. The observer's sight was reflected by the mirror at the magnet on a scale. Already during the 19[th] century the photographic recording was constructed. A light beam from a lamp is reflected by the mirror at the magnet on a photographic paper, which is fixed on a drum, driven by a clock work, which takes one turn per day. The photographic paper was to be changed once per day and developed in the normal way. The product was called magnetogram. The variometer rooms needed to be kept in complete darkness. Only for the short time of paper change week red light was allowed.

In the beginning of photographic recording the light sources were gasoline lamps. They were in use in Potsdam and Seddin. In Niemegk electrical lamps, powered by batteries, were in use. In Seddin and Niemegk Helmholtz coils, mounted at every variometer were in use for scale value determination. By means of glass scales the positions of the tracks at the magnetograms were read and converted using the scale values into magnetic units. Comprehensive details on classical instruments and measurement practices can be found in (Wiese et al., 1960b).

Absolute measurements are performed by using magnetic theodolites. By means of them the declination (D) and the horizontal intensity (H) can be determined. They consist of a horizontal circle, a telescope, a magnet (the so called magnetic needle) and a vertical thread. It is placed on a stable pillar. It is necessary to know the geodetical azimuth of the pillar with respect to an azimuth mark. The geodetical azimuth is the angel between the geographic North direction and the sight line from the measurement pillar to the azimuth mark. The

declination measurement starts with a bearing by means of the telescope to the azimuth mark and the reading of the circle. Then the magnetic needle is fixed at the vertical thread. The natural Earth's magnetic field will force the magnetic needle to point to magnetic North. The observer needs to turn the theodolite, until the magnetic needle is parallel to the axis of the telescope, by observing the face side of the magnetic needle. Now the horizontal circle needs to be read. The difference of both the readings is the declination.

For the measurement of the horizontal intensity the theodolite needs to have one or two bars at the same level as the magnetic needle is placed. A so called deflection magnet of a known magnetic force is put on the bar in a fixed distance perpendicular to the magnetic needle. From the angular difference of a bearing with and without the deflection magnet H can be calculated. A second method of H measurement is the oscillation method. The deflection magnet is fixed in its centre at a vertical thread and forced to make torsion movements. The horizontal intensity can be calculated from the oscillation period. The performance of 2 independent methods for the same parameter allows to eliminate drifts in the magnetic force of the deflection magnet. The determination of the oscillation period can be done by the "eye and ear" method. More accurate results yield the use of a suitable opto-electronic equipment (Schmidt, 1956).

The inclination (I) was in ancient times determined by a dip needle, which is a magnetic needle, which may move in vertical direction. The position of the needle with respect to a vertical circle gives the value of I. A more precise instrument is the earth inductor. It uses a zero-method. The position of a rotating coil is to be found, in which no current in the coil is induced caused by the Earth's magnetic field. A sensitive galvanometer is connected to the coil as the indicator for the induced current. Tilt and azimuth of the coil axis can be adjusted and read on reference circles. The read tilt is the inclination.

Beginning with the 1940s new measurement methods and instruments for the Earth's magnetic field were developed. Around 1990 instruments using the new principles were able to be used in the magnetic observatories. Three principles found wide application in the observatory practice:

Coils of many windings without cores came in use for the recording of short periodic variations. The voltages induced in the coils were recorded on photographic paper by galvanometers. Later on coils with high permeable cores (search coil magnetometers) were in use with digital recording.

A further principle is the saturation core or flux-gate magnetometer. The working principle of this instrument is based on the saturation of a transformer core. In this situation primarily odd harmonics of the excitation frequency are produced. When in addition to the alternating excitation field also a DC component of the environmental magnetic field acts on the core, also even harmonic signals appear. By means of suitable electronic circuits these even harmonics are detected and converted into an output voltage proportional to the Earth's magnetic field component in the direction of the sensor core.

A third principle is the so called proton magnetometer. It consists of a vessel filled with a proton-rich liquid, a surrounding coil, and electronic control circuits. The coil is used in two modes: In the first one a strong DC current flows through the coil. The induced magnetic field forces the protons in the liquid sample to align their spin axis along the coil field. After switch off of the DC current the protons start to reorient their spin axis towards the ambient magnetic field by performing a precession motion. The resulting precession frequency is proportional to the ambient magnetic field strength. In this phase the coil is used to pick up the proton precession signal and the electronics measures the derived frequency. Best results are achieved when the direction of the applied DC field is approximately perpendicular to the Earth's magnetic field.

Further principles were developed, but their use in magnetic observatories is limited. Only the optical pumping is some useful, which is based on an atomic effect of the electrons. The construction of such magnetometers is extremely expensive (Bloom, 1962; Pulz and Jäckel, 1998).

Widely in use for absolute measurements are DI-flux magnetometers, which consist of a non-magnetic theodolite, having a horizontal and a vertical circle and is equipped by a fluxgate sensor, connected to its electronic unit. The axis of the fluxgate is adjusted as good as possible parallel to the telescope axis and serves as an indicator for the direction of the Earth's magnetic field. It is possible to determine the declination (D) and the inclination (I). The instrument is placed on a stable pillar. It is necessary to know the geodetical azimuth of the pillar with respect to an azimuth mark. The geodetical azimuth is the angel between the geographic North direction and the sight line from the measurement pillar to the azimuth mark. The declination measurement starts with a bearing by means of the telescope to the azimuth mark and the reading of the circle. The next step is to align the telescope as exactly as possible horizontally. Then the telescope is rotated until the display of the magnetometer

shows zero. That means, the direction of the Earth's magnetic field in the horizontal plane is perpendicular to the telescope. The horizontal circle is to be read. The difference of both the readings, considering the azimuth value and 90° is the declination. The inclination can be determined by tilting the telescope until the display of the magnetometer shows zero. Reading the vertical circle, considering 90° (because the position of the telescope is perpendicular to the direction of the Earth's magnetic field in the vertical plane). Both angles are determined by means of a zero method, which eliminates offset errors and non-linearities of the magnetometer. The misalignments of the sensor axis with respect to the telescope axis are eliminated by means of performing the D and I measurements in 4 positions each.

The total intensity (F) of the Earth's magnetic field is determined by means of a proton magnetometer. The 2 angles, D and I and the scalar value F determine completely the field vector. The natural variation of the field during the time of the absolute measurement is taken into account by considering the variometer data. The instruments and the procedure are described in detail in (Jankowski and Sucksdorff, 1996).

## Appendix II – Further Observatory Equipment

On the occasion of the International Polar Year 1932/1933 an equipment for recording of telluric currents was installed, consisting of 2 lines in the geographic directions North-South and East-West, both of 1,000 m length, funded by the Rockefeller foundation (Bock, 1950). The electrodes were lead plates. The East and South electrodes were located at the observatory compound, the West and North electrodes in the appropriate distance in neighbouring forests. The recording was performed by galvanometers on photographic paper, located in the laboratory. It is unknown, when the recordings were stopped; recordings are not anymore available. In 1949 activities for the re-establishment of telluric recordings were started. Potential-free copper – blue vitriol electrodes were developed in 1953 at the observatory (Lengning, 1958). In 1956 the continuous recording was started (Lengning, 1960). From 1957 onward further lines of different directions and lengths were installed and operated. Presently only the two 1,000 m geographically oriented lines are still in operation, using digital recording since 2001 (Linthe and Schulz, 2007).

In 1952 induction coil variometers were installed and operated. Rectangular coils of many windings and big dimensions located in the absolute house were in use for the North and East component. The vertical intensity was detected by a wire fixed at the fence around the observatory compound. The coils were connected to galvanometers recording on

photographic paper of a speed of 4 mm per minute in the South room of the variation house (Wiese, 1956). Due to the enlargement of the observatory compound in 1952 a horizontal coil of 50 m circumference was installed as vertical intensity sensor (Wiese, 1958). The recording galvanometer was moved in 1957 to the West room of the variation house (Wiese, 1960a). In 1971 the coreless coils were replaced by cylindrical ones with cores of high permeability of small diameter and 2 m length (search coils). An electronic amplifier unit, developed during the period 1965-1970 with photographic recording was taken into operation (Auster, 1972). The photographic recording was in 1999 replaced by digital one. The equipment is still in operation.

## Appendix III – Activities of the Observatory, Exceeding its Ordinary Purpose

Around 1948 a field balance on the base of a tape-suspended magnet was constructed in the precise mechanic workshop of the observatory (Fanselau, 1948). This instrument improved dramatically the knife-edge field balance after Adolf Schmidt. In the beginning of the 1950s the development of instruments on the base of new principles was started: flux-gate and proton magnetometers. The tape-suspended field balance was elaborated more and more, different modifications were constructed. The project of constructing a chamber of constant magnetic field ("Konstanthaltung") for instrument calibration was started (Fanselau, 1953). In 1952 the compound of the observatory was enlarged to a size of 5.2 ha to ensure the undisturbed operation of the Earth magnetic observations besides the further projects of instrumental development. A measurement and adjustment hut for the field balance production and 2 huts for the constant magnetic field chamber (one containing a 3-component cylindrical Helmholtz coil system of big dimensions) were constructed (Fanselau, 1955).

In preparation of the International Geophysical Year (IGY) 1957 in 1953 three satellite stations were started to be constructed in Warnkenhagen (North-West German Democratic Republic), Ückermünde (North-East GDR) and Herrnhut (South-East GDR) for geomagnetic and geoelectric recordings (Fanselau, 1956). The data of the satellite stations were intended to be used for scientific investigations of the distribution of the Earth's magnetic field over the GDR and local differences of the secular variation. The Herrnhut station was terminated in 1961 (Fanselau, 1962b). The Ückermünde station existed until 1965 (Fanselau, 1966). The closure of the 2 stations happened due to logistic and financial reasons.

In 1956 the precision mechanical workshop was moved from the basement of the main building to the storage house, suitably modified for this purpose. The instrumental equipment of the satellite stations was continued in 1956 (Fanselau, 1957).

In 1956 a project to study the local gradient of the Earth magnetic field was started. For this purpose 4 magnetometer stations were constructed at the corner points of a 7 km square, geographically oriented (Fanselau, 1958). Each station was equipped with 3 geographically oriented photo-electrically compensated field balances with analogue paper recording. The north-western station was located in a distance of 200 m south-westward of the observatory compound.

Different measurement expeditions were performed in connection with the IGY: repeat station and magneto-telluric measurements at the territory of the GDR and some Eastern European countries from 1956 onward. In 1961 an expedition to study the effect of a solar eclipse on the Earth's magnetic field in Romania and Bulgaria took place (Fanselau, 1962a). A van "Phänomen Granit 30 K" (Fig. 24 shows a photo of it) was in use for all expeditions. The van was completely equipped with any necessary instruments.

From 1953-1962 repeat station measurements on 1762 stations on the territory of the German Democratic Republic were carried out. The results were reduced on the period 1957.5 (Bolz, et al., 1969).

In 1963 electronic data processing started by means of purchase of a small computer Cellatron SER 2, produced by the company Cellatron in Zella-Mehlis, Thuringia, German Democratic Republic (Fanselau, 1964). An equipment for digitization of photographic magnetic recordings was developed and taken into use together with the small computer. The yearbook 1965 was the first one produced by means of the use of the SER 2 (Schmidt, 1967).

From 1970 onward the observation program of the remote station Warnkenhagen was enlarged. Besides the variometer set of sensitive scale values a set of lower sensitivity and a scalar proton magnetometer was installed. Also telluric and induction coil magnetometer recordings were taken into operation (Lengning et al., 1973). From 1976 onward a vector proton magnetometer was in operation. At a further remote station, located at Sosa in the Erzgebirge, a scalar proton magnetometer was installed in 1978.

In 1972 a digital data acquisition equipment based on modules of the computer ROBOTRON R300, produced by the company Robotron in Radeberg, near Dresden, German Democratic

Republic was installed for the recording of 1 Hz- sampled fife canals magnetic and telluric data in the enlarged computer house (building No. 3 at Fig. 15) of the observatory (Lengning et al., 1973).

In 1975 a process control computer PRS4000, produced as well as by the company Robotron, was installed in the again enlarged computer house of the observatory (Lengning et al., 1976). It was intended to be used for the direct digital data acquisition from the geomagnetic recording instruments and for data processing. It was in operation during the 3 regular world days for the on-line processing of the signals of the search coil magnetometers.  From 1976 onward the yearbook tables were produced by means of this computer. All the digital proton magnetometer recordings on punched tapes were inserted and stored on the PRS4000.

In 1983 a microcomputer MPS4944 was taken into operation for the continuous on-line processing of the signals of the search coil magnetometers. The MPS4944 was produced by the scientific toolbuilding of the Central Institute for Nuclear Research in Rossendorf near Dresden, German Democratic Republic, which is today the Helmholtz-Zentrum Dresdan-Rossendorf. The necessary software was developed at the observatory (Lenners et al., 1984).

The remote stations Warnkenhagen at the Baltic Sea coast as well as Sosa at Erzgebirge mountains were closed in June 1991 due to the changed conditions caused by the German reunification in 1990.

On the base of contributions of the observatories Fürstenfeldbruck, Niemegk and Wingst the first entire German magnetic map after the German reunification was published in 1995 (Beblo et al. 1995).

The self-made vector proton magnetometer on pillar No. 1 in the absolute house was in operation until 1998, but its results were never in use for the observatory data (Linthe, 2000).

A long tradition of instrument development existed at the Niemegk Adolf Schmidt Geomagnetic Observatory. The precise mechanical workshop and the electronic laboratory ensured excellent conditions. The staff reduction of the observatory caused by the German reunification decreased the instrumental development base. But several activities took place further. Eberhard Pulz intensively worked on the design of optically pumped magnetometers, using international experiences. He designed for instance a caesium-potassium tandem magnetometer (Pulz and Jäckel, 1998). Also on the base of his international contacts a caesium-helium magnetometer was built (Pulz and Linthe, 1998). The scalar caesium-

potassium magnetometer is still in test operation on pillar No. 15 of the absolute house. On the base of the experience with these scalar instruments later on a vector magnetometer was developed (Pulz, et al., 2009). The sensor of the instrument is placed in the small hut (house No. 9 at Fig. 16). The instrument is still in permanent test operation.

The Institute of Geophysics and Extraterrestrial Physics of the Technical University Brunswick proposed the development of an automatic absolute measurement system on the base of the satellite magnetometer calibration experiences (Auster et al., 2007, Hemshorn et al. 2009). The precise mechanical workshop of the Niemegk Adolf Schmidt Geomagnetic Observatory manufactured the hardware and installed it on pillar No. 5 in the absolute house. The software was designed by the Brunswick and Niemegk teams. The instrument is still in permanent test operation.

Julius Bartels suggested in 1949 to the international geomagnetic community to establish an international service of producing a general index, which describes the geomagnetic planetary activity, caused by solar corpuscular radiation. He called this activity index Kp "planetarische Kennziffer" (Bartels, 1949). His suggestion was accepted by the international community and was continuously produced under his leadership at the Geophysical Institute of the Göttingen University. After Bartels' death in 1964 his successor Manfred Siebert continued the service. Considering his retirement Siebert asked for any responsible successor institution within the Deutsche Geophysikalische Gesellschaft (German Geophysical Society). The Niemegk Adolf Schmidt Geomagnetic Observatory was ready to take over the service. Therefore, on 1 January 1997 the Kp Index Service of the International Service of Geomagnetic Indices was taken over from the Geophysical Institute of the Göttingen University (Best and Linthe, 1999). The index is of global relevance and did not lose any significance and usage during the years.

**Appendix IV - Selection of Significant Meetings and Conferences Related to the Observatory**

Several scientific and technical conferences were held from time to time at the observatory.

On 11 and 12 November 1960 a commemorate event was held at the Humboldt University Berlin, honouring Adolf Schmidt's 100[th] birthday on 23 July 1960 and the 150[th] anniversary of the university. A further memorial on the occasion of the 30th anniversary of the observatory took place on 21 December 1960 (Fanselau, 1962a).

On the occasion of the 50 years anniversary of the Niemegk Adolf Schmidt Geomagnetic Observatory the international symposium "Current problems of the geomagnetic research" took place at the observatory and at a holiday camp in 20 km distance. Almost 100 participants attended the symposium. About 50 scientific presentations were given and 13 participants performed comparison measurements by means of their own instruments (Kautzleben, 1981).

The Central Institutes for Physics of the Earth and Solar-Terrestrial Physics performed on 29 April 1983 in Potsdam a colloquium honouring Gerhard Fanselau, former director of the Geomagnetic Institute Potsdam and Niemegk Adolf Schmidt Geomagnetic Observatory. Seven scientific presentations were given (Lengning et al., 1983).

The Heinrich Hertz Institute for Atmosphere Research and Geomagnetism Berlin performed 22-26 September 1986 the IAGA Symposium on Space-Time-Structure of the Geomagnetic Field in Lutherstadt Wittenberg including a visit of the Niemegk Adolf Schmidt Geomagnetic Observatory (Mundt, 1987).

From 23-28 April 1990 the International Symposium 100 Years Geomagnetic Observatory Potsdam – Seddin – Niemegk was held in Potsdam. Sixty scientists from 14 countries participated. Forty four scientific presentations were given. The participants visited the historic magnetic measurement buildings on Telegrafenberg Potsdam and the Niemegk Adolf Schmidt Geomagnetic Observatory (Mundt and Best, 1991; Best et al., 1991; Best et al., 1992).

On 7 an 8 September 1996 the INTERMAGNET Executive Council and Operations Committee held a meeting at the Niemegk Adolf Schmidt Geomagnetic Observatory.

The observatory organized the VII[th] IAGA Workshop on Geomagnetic Instruments and Data Acquisition from 9-14 September 1996. Ninety fife scientists from 33 countries participated in the workshop. During the practical part 45 absolute measurements were performed at the observatory. During the scientific part 48 papers and 12 posters were presented in the ALBA Hotel Wittenberg. The results and papers were published (Best and Linthe, 1998).

In collaboration with the German Esperanto League a commemoration on Adolf Schmidt's 50[th] year of death took place at the observatory on 17 October 1994 (Best and Wollenberg 1994).

On 23 July 2010 Adolf Schmidt's 150[th] birthday and the 80 years anniversary of the opening of the Niemegk Adolf Schmidt Geomagnetic Observatory were celebrated in Niemegk by the Deutsche Geophysikalische Gesellschaft and the Helmholtz Centre Potsdam (Jacobs and Linthe, 2010). Thirty participants attended the festivity.

**Appendix V –Collaboration with International Observatories**

The Adolf Schmidt Niemegk Geomagnetic Observatory maintained a closed collaboration with many international geomagnetic observatories. Scientific mutual visits took place in a big number. Comparison measurements were carried out at Niemegk and international observatories to compare the accuracy of the instruments and the observers.

In 1996 the Hurbanovo Geomagnetic Observatory (Slovakia) was supported by providing new instruments for the absolute measurements and variation recordings funded by the Volkswagen foundation. The Hurbanovo staff was trained in the use of the instruments by Hans-Joachim Linthe.

From 2005 onward new geomagnetic observatories were established or existing ones equipped with modern instruments on the base of international agreements, sponsored by Helmholtz Centre Potsdam - GFZ. The observatories are listed in table 1.

Meetings of German speaking observers were held from time to time in Niemegk or at other observatories. Students education of several German universities is supported in the frame of excursions and practical training. Guided tours through the observatory are offered to any interested persons.

Since 1961 the observatory instructs regularly trainees in precise mechanics and electronics.

The agency for military surveying of the German Federal Armed Forces regularly calibrated their magnetic instruments at the observatory and took consult on magnetic measurement instruments and measurement practice.

**Appendix VI - Internationally Awarded Employees of the Observatory**

Walter Zander (1922-1998) was awarded with the Long Service Award of the International Association of Geomagnetism and Aeronomy (IAGA) in 1993 for his outstanding contribution to produce high quality data by the Niemegk Adolf Schmidt Geomagnetic Observatory (IAGA News 1993). Fig. 25 shows the handing over of the medal to Walter

Zander by Heinrich Soffel (National Representative of Germany for IAGA). The same award was presented to Hans-Joachim Linthe in 2015 for his dedicated effort for the operation of the Niemegk Adolf Schmidt Geomagnetic Observatory and the modernization or new establishment of international observatories (Mandea, 2014). He was further an active member of the INTERMAGNET Operations Committee (2003-2014) and chair of the Working Group V-OBS of the IAGA (2003-2007). At Fig. 26 Linthe (right) is to be seen together with Kathy Whaler (IAGA President 2011-2015, left) and Mioara Mandea (IAGA Secretary General 2009-2019).

**Appendix VII - Prominent Scientific Results and Instrumental Achievements Connected with the Observatories Potsdam – Seddin – Niemegk**

Max Eschenhagen: Classification of days into 5 categories regarding the magnetic activity (Eschenhagen, 1894); introduction of the "Gamma, γ" as a unit in geomagnetism (Eschenhagen, 1896); Discovery of pulsations "Elementarwellen" (elementary waves), (Eschenhagen, 1897).

Adolf Schmidt: Calculation of the geomagnetic potential for the epoch 1885 (Schmidt, 1885); transformation of spherical harmonics into different coordinate systems (Schmidt, 1889); construction of the knife edge field balance (Schmidt, 1915); simplification of Eschenhagen's classification of days by introduction of 3 categories, introduction of the International Character Figure Ci and the inter-diurnal variability (Schmidt, 1916); construction of a new magnetic theodolite for an improved method of the deflection experiment (Bock and Schmidt, 1928).

Julius Bartels: Introduction of the activity index K "Kennziffer" in 1939 from Niemegk recordings (Bartels, 1938); introduction of the planetary activity index Kp "planetarische Kennziffer" and derived indices ap, Ap, Cp and C9, the internationally most used measure to characterise geomagnetic activity (Bartels, 1949).

Richard Bock: "Magnetische Reichsaufnahme" – repeat station campaign over Germany "Deutsches Reich" together with F. Burmeister and F. Errulat (Bock, Burmeister and Errulat, 1948); high merits in the changeover of the observation service from Potsdam and Seddin to Niemegk (Bock, 1950).

Gerhard Fanselau: Improvement of the field balance by using a suspended balance (Fanselau, 1948).

Horst Wiese: Discovery of the North German conductivity anomaly together with O. Meyer

(Wingst) – basement of his theoretical contributions to magneto-tellurics – "Wiese Arrow"

(Wiese, 1965).

H. Schmidt: Construction of several observatory instruments: proton magnetometers,

fluxgates, induction coil variometers (Schmidt, 1962; Wiese, 1962); introduction of data

processing into the observatory practice (Fanselau, 1964).

**Competing Interests**

I declare that I do not have any conflict of interest.

**Acknowledgements**

Kristian Schlegel strongly encouraged me to write this paper. I am very thankful to him for

his patience. I further thank the Helmholtz Centre Potsdam GFZ German Research Centre for

Geosciences, which rendered possible to write the paper using its resources. I am especially

thankful to Jürgen Matzka, group leader Geomagnetic Observatories of the Geomagnetism

Section of the GFZ, for giving me the opportunity to work at the Niemegk Adolf Schmidt

Geomagnetic Observatory. Since my official retirement end of 2014 I had the chance to use

an office, a computer and all the observatory publications to collect the necessary

information.

| Year | IAGA code | Name | Country |
|------|-----------|------|---------|
| 2005 | PAG | Panagyurishte | Bulgaria |
| 2005 | KMH | Keetmanshoop | Namibia |
| 2006 | YAK | Yakutsk | Russia |
| 2007 | MGD | Magadan | Russia |
| 2007 | SHE | St. Helena | British Overseas Territory |
| 2007 | ABG | Alibag | India |
| 2009 | SUA | Surlari | Romania |
| 2009 | PET | Paratunka | Russia |
| 2010 | SMA | Santa Maria | Portugal – Azores |

| 2013 | ODE | Odessa | Ukraine |
|------|-----|--------|---------|
| 2014 | VSS | Vassouras | Brazil |
| 2014 | TDC | Tristan da Cunha | British Overseas Territory |
| 2015 | TTB | Tatuoka | Brazil |
| 2015 | BFO | Black Forrest | Germany |
| 2015 | VNA | Neumayer Station III | Germany's Antarctic Station |
| 2018 | GAN | Gan | the Maldives |
| 2019 | STT | Sao Teotonino | Portugal – Azores |
| 2022 | LRV | Leivogur | Iceland |

1    Table 1. List of the observatories, newly established or equipped with modern instruments on

2    the base of the international collaboration with Helmholtz Centre Potsdam – GFZ, Niemegk

3    Adolf Schmidt Geomagnetic Observatory.

| Time period | Institutional Affiliations |
|-------------|----------------------------|
| 1932-1933 | Magnetic department of the Magnetic Meteorological Observatory Potsdam of the Prussian Meteorological Institute Berlin |
| 1934-1936 | Magnetic Observatory of the Berlin University in Potsdam-Niemegk |
| 1937-1950 | Geophysical Institute Potsdam |
| 1950-1956 | Geomagnetic Institute and Observatory Potsdam/Niemegk of the Meteorological and Hydrological Service of the Interior Ministry of the German Democratic Republic |
| 1957-1968 | Geomagnetic Institute and Observatory Potsdam-Niemegk of the German Academy of Sciences Berlin |
| 1969-1981 | Central Institute for Physics of the Earth Potsdam |
| 1982-1983 | Central Institute for Solar-Terrestrial Physics Berlin |
| 1984-1991 | Heinrich Hertz Institute for Geomagnetism and Atmosphere Research Berlin |

| From 1992 onward | GeoForschungsZentrum Potsdam, in 2008 renamed into Helmholtz Centre Potsdam GFZ German Research Centre for Geosciences |
|---|---|

Table 2. Affiliations of the Niemegk Adolf Schmidt Geomagnetic Observatory

| Time period | Scientific Directors resp. Heads | Portrait |
|---|---|---|
| 1932-1936 | Alfred Nippoldt (1874-1936) | Fig. 13 in Linthe, 2023a |
| 1937-1945 | Julius Bartels (1899-1964) | Fig. 27, left |
| 1945-1969 | Gerhard Fanselau (1904-1982) | Fig. 28, left |
| 1969-1982 | Herbert Schmidt (1921-1981), Klaus Lengning (1917-2000) | Fig. 29, left Fig. 29, right |
| 1983-1998 | Adolf Best (1933-2012) | Fig. 30, right |
| 1999-2001 | Richard Holme (born in 1967) | Fig. 31, right |
| 2002-2014 | Monika Korte (born in 1971) | Fig. 31, left |
| From 2014 onward | Jürgen Matzka (born in 1971) | Fig. 33 |

Table 3. Scientific directors resp. heads of the Niemegk Adolf Schmidt Geomagnetic Observatory

| Time period | Observers | Portrait |
|---|---|---|
| 1932-1933 | Richard Bock (1899-1961) | Fig. 27, right |
| 1934-1951 | Gerhard Fanselau (1904-1982) | Fig. 28, left |
| 1952-1961 | Horst Wiese (1922-1972) | Fig. 28, right |
| 1962-1968 | Armin Grafe (born in 1934) | Fig. 30, left |
| 1969-1982 | Klaus Lengning (1917-2000) | Fig. 29, right |
| 1983-1991 | Eberhard Ritter (born in 1934) | Fig. 32, left |
| 1992-2014 | Hans-Joachim Linthe (born in 1949) | Fig. 32, right |
| From 2014 onward | Jürgen Matzka (born in 1971) | Fig. 33 |

1    Table 4. Observers of the Niemegk Adolf Schmidt Geomagnetic Observatory

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

Fanselau, G.: Erwähnenswertes aus dem Geomagnetischen Institut und Adolf-Schmidt-Observatorium für Erdmagnetismus, Jahrbuch 1964 des Adolf-Schmidt-Observatoriums für Erdmagnetismus in Niemegk, Akademie-Verlag, Berlin 1966

Fanselau, G.: Erwähnenswertes aus dem Geomagnetischen Institut und Adolf-Schmidt-Observatorium für Erdmagnetismus (1.1.1967 bis 31.12.1967), Jahrbuch 1966 des Adolf-Schmidt-Observatoriums für Erdmagnetismus in Niemegk, Akademie-Verlag, Berlin 1968

Fanselau, G.: Erwähnenswertes aus dem Geomagnetischen Institut und Adolf-Schmidt-Observatorium für Erdmagnetismus, Jahrbuch 1967 des Adolf-Schmidt-Observatoriums für Erdmagnetismus in Niemegk, Akademie-Verlag, Berlin 1969

Förster, H.: Genaue Azimutbestimmung auf einem Pfeiler des Adolf-Schmidt-Observatoriums in Niemegk – einer Außenstelle des GeoForschungsZentrums Potsdam, Diplomarbeit am Institut für Planetare Geodäsie der Technischen Universität Dresden, 1998

Grafe, A.: Ergebnisse der Beobachtungen am Adolf-Schmidt-Observatorium für Erdmagnetismus in Niemegk im Jahre 1966, Jahrbuch 1966 des Adolf-Schmidt-Observatoriums für Erdmagnetismus in Niemegk, Akademie-Verlag, Berlin 1968

Hemshorn, A., Pulz, E., Mandea, M.: GAUSS: Improvements to the Geomagnetic Automated SyStem, Proceedings of the XIII[th] IAGA Workshop on Geomagnetic Observatory Instruments, Data Acquisition and Processing, pp. 100-103, U.S. Geological Survey Open-File Report 2009-1226, Boulder and Golden, Colorado, U.S.A., 2009

IAGA News No. 32, Published by the Secretary General of IAGA, Fraser Noble Building, Aberdeen University, Aberdeen AB9 2UE, Scotland, (UK), Printed by Woods of Perth (Printers) Ltd., 1993

Jacobs, F., Linthe, H.-J.: 150 Jahre Adolf Schmidt und 80 Jahre Observatorium Niemegk, pp. 46-48, DGG-Mitteilungen, Potsdam,  3/2010, ISSN 0934-6554

Jankowski, J., Sucksdorff, C.: IAGA Guide for Magnetic Measurements and Observatory Practice, ISBN: 0-9650686-2-5, Warsaw, 1996

Kautzleben, H.: Vorwort; 90 Jahre Adolf-Schmidt-Observatorium für Erdmagnetismus in Niemegk – 90 Jahre geomagnetische Forschung in Potsdam, pp. 5-23, Veröffentlichungen des Zentralinstituts für Physik der Erde Nr. 70 Teil 1, Potsdam,  1981

Lengning, K.: Die Erdstromapparatur am Observatorium in Niemegk, pp. 160-165, Jahrbuch 1955 des Adolf-Schmidt-Observatoriums für Erdmagnetismus in Niemegk, Akademie-Verlag Berlin, 1958

Lengning, K.: Angaben zur Erdstrom-Registrierung, pp. 45-48, Jahrbuch 1957 des Adolf-Schmidt-Observatoriums für Erdmagnetismus in Niemegk, Akademie-Verlag Berlin, 1960

Lengning, K., Schmidt, H., Zander, W.:  Jahrbuch 1970 des Adolf-Schmidt-Observatoriums für Erdmagnetismus in Niemegk, Zentralinstitut Physik der Erde  Potsdam, 1971

Lengning, K., Schmidt, H., Zander, W.:  Jahrbuch 1972 des Adolf-Schmidt-Observatoriums für Erdmagnetismus in Niemegk, Zentralinstitut Physik der Erde  Potsdam, 1973

Lengning, K., Schmidt, H., Zander, W.:  Jahrbuch 1975 des Adolf-Schmidt-Observatoriums für Erdmagnetismus in Niemegk, Zentralinstitut Physik der Erde  Potsdam, 1976

Lengning, K., Schmidt, H., Zander, W.:  Jahrbuch 1976 des Adolf-Schmidt-Observatoriums für Erdmagnetismus in Niemegk, Zentralinstitut für Solar-Terrestrische Physik, 1977

Lengning, K., Lenners, D., Zander, W.:  Jahrbuch 1982 des Adolf-Schmidt-Observatoriums für Erdmagnetismus in Niemegk, Zentralinstitut Physik der Erde  Potsdam, 1983, ISSN 0065 - 2016

Lenners, D., Ritter, E., Zander, W.:  Jahrbuch 1983 des Adolf-Schmidt-Observatoriums für Erdmagnetismus in Niemegk, Zentralinstitut Physik der Erde Potsdam, 1984, ISSN 0065 - 2016

Linthe, H.-J.: Bestimmung der Pfeilerdifferenzen in der Totalintensität im Absoluthaus, Jahrbuch 1994 des Adolf-Schmidt-Observatoriums für Geomagnetismus in Niemegk, GeoForschungsZentrum Potsdam, 1995, ISSN 0065 - 2016

Linthe, H.-J.: Jahrbuch 1996 des Adolf-Schmidt-Observatoriums für Geomagnetismus in Niemegk, GeoForschungsZentrum Potsdam, 1997

Linthe, H.-J.: Jahrbuch 1998 des Adolf-Schmidt-Observatoriums für Geomagnetismus in Niemegk, GeoForschungsZentrum Potsdam, 2000, ISSN 0065 - 2016

Linthe, H.-J., Schulz, Günter: Yearbook Magnetic Results 2000, Adolf Schmidt Geomagnetic Observatory Niemegk, Geomagnetic Observatory Wingst, GeoForschungsZentrum Potsdam, 2005, ISSN 1614-5801, DOI: 10.2312/GFZ.b103-20004 ; urn:nbn:de:kobv:b103-20004

Linthe, H.-J., Schulz, Günter: Yearbook Magnetic Results 2001, 2002, 2003 Adolf Schmidt Geomagnetic Observatory Niemegk, Geomagnetic Observatory Wingst, GeoForschungsZentrum Potsdam, 2007, ISSN 1614-5801, DOI: 10.2312/GFZ.b103-yb2001_20037

Linthe, H.-J.: History of the Potsdam, Seddin and Niemegk Geomagnetic Observatories – First Part: Potsdam, Hist. Geo Space. Sci., 14, 23–31, https://doi.org/10.5194/hgss-14-23-2023, 2023a

Linthe, H.-J.: History of the Potsdam, Seddin and Niemegk Geomagnetic Observatories – Second Part: Seddin, Hist. Geo Space. Sci., 14, 43–50, https://doi.org/10.5194/hgss-14-43-2023, 2023b

Luyken, K.: Der Pantograph für Registrier-Kurven von Ad. Schmidt (Potsdam), Zeitschrift für Instrumentenkunde XXIX, Januar 1909, Erstes Heft, pp. 1-14

Mandea, M. (Executive editor): IAGA News No. 52, GFZ German Research Centre for Geosciences, Potsdam 2015

Mundt, W.: HHI Report No. 21, Proceedings of the IAGA Symposium Space-Time-Structure of the Geomagnetic Field, September 22-26, 1986 in Wittenberg, GDR, Berlin, 1987, ISSN 0863-0607

Mundt, W., Best, A.: HHI Report No. 22, Proceedings of the International Symposium 100 Years Geomagnetic Observatory Potsdam – Seddin – Niemegk, April 23-28, 1990 in Potsdam, Berlin, 1991, ISSN 0863-0607

A. Nippoldt: Verlegung der magnetischen Observatorien von Potsdam und Seddin wegen Elektrisierung der Vorortbahnen, pp. 52-60, Bericht über die Tätigkeit des Preußischen

Meteorologischen Instituts im Jahre 1928, Veröffentlichung des Preußischen Meteorologischen Instituts Nr. 362, Berlin, 1929

Nippoldt, A.: Ergebnisse der magnetischen Beobachtungen in Potsdam und Seddin im Jahre 1928, pp. 3-38, Veröffentlichungen des Preußischen Meteorologischen Instituts Nr. 374, Julius Springer, Berlin, 1930

Nippoldt, A.: Magnetische Arbeiten, In: Bericht über die Tätigkeit des Preußischen Meteorologischen Instituts im Jahre 1929 erstattet vom Direktor, pp. 35-39, Veröffentlichung des Preußischen Meteorologischen Instituts Nr. 372, Julius Springer Berlin, 1930

Nippoldt, A.: Magnetische Arbeiten, In: Bericht über die Tätigkeit des Preußischen Meteorologischen Instituts im Jahre 1930 erstattet vom Direktor, pp. 44-49, Veröffentlichung des Preußischen Meteorologischen Instituts Nr. 380, Behrend & Co. Berlin, 1931a

Nippoldt, A.: Ergebnisse der magnetischen Beobachtungen in Potsdam und Seddin im Jahre 1929, pp. 3-35, Veröffentlichungen des Preußischen Meteorologischen Instituts Nr. 383, Julius Springer, Berlin, 1931b

Nippoldt, A.: Ergebnisse der magnetischen Beobachtungen in Potsdam und Seddin im Jahre 1931, Magnetisches Observatorium der Universität Berlin in Potsdam-Niemegk, Julius Springer, pp. 3-52, Berlin, 1934

Pulz, E., Jäckel, K.-H.: A new design for an optically pumped tandem magnetometer, Proceedings of the VII[th] IAGA Workshop on Geomagnetic Observatory Instruments, Data Acquisition and Processing, GeoForschungsZentrum Potsdam Scientific Technical Report STR98/21, pp. 155-167, Potsdam, 1998

Pulz, E., Linthe, H.-J.: A Long term investigation of two different optically pumped scalar magnetometers, Proceedings of the VII[th] IAGA Workshop on Geomagnetic Observatory Instruments, Data Acquisition and Processing, GeoForschungsZentrum Potsdam Scientific Technical Report STR98/21, pp. 136-146, Potsdam, 1998

Pulz, E., Jäckel, K.-H., Bronkalla, O.: A quasi absolute optically pumped magnetometer fort he permanent recording of the Earth's magnetic field vector, Proceedings of the XIII[th] IAGA Workshop on Geomagnetic Observatory Instruments, Data Acquisition and Processing, pp. 216-219, U.S. Geological Survey Open-File Report 2009-1226, Boulder and Golden, Colorado, U.S.A., 2009

Richard, M., Wiese, H.: Die Neubestimmung der absoluten erdmagnetischen Feldgrößen am Adolf-Schmidt-Observatorium für Erdmagnetismus in Niemegk, Abhandlung Nr. 13 des Geophysikalischen Instituts Potsdam, Akademie-Verlag Berlin, 1954

Schmidt, A.: Mitteilungen über eine neue Berechnung des erdmagnetischen Potentials. Abh. Bayer. Akad. D. Wiss., II. Klasse, 19, 1895, pp. 1–66.

Schmidt, A.: Mathematische Entwicklungen zur allgemeinen Theorie des Erdmagnetismus. In: Aus dem Archiv der Deutschen Seewarte. 12. Jg. Nr. 3, 1889.

Schmidt, A.: Ein Lokalvariometer für die Vertikalintensität. Ber. Tätigkeit Kgl. Preuß. Meteorol. Inst, im Jahre 1914, pp. 109–134, Berlin 1915

Schmidt, A.: Die internationalen erdmagnetischen Charakterzahlen. Meteor. Z. 33 (1916), S. 481-492

Schmidt, A.: Eine photographische Registriereinrichtung mit weiter Zeitskala bei sparsamem Papierverbrauch, pp. 38-45, Bericht über die Tätigkeit des Preußischen Meteorologischen Instituts im Jahre 1925 erstattet vom Direktor, Veröffentlichung des Preußischen Meteorologischen Instituts Nr. 335, Behrend & Co., Berlin 1926

Schmidt, H.: Untersuchungen zur Theorie und Praxis geomagnetischer Schwingungs-messungen mit Beschreibung einer neuen Schwingzeitmeßanlage, Abhandlung Nr. 19 des Geomagnetischen Instituts und Observatoriums Potsdam-Niemegk, Akademie-Verlag Berlin, 1956

Schmidt, H.: Beobachtung der Totalintensität mit Protonenmagnetometern, Jahrbuch 1959 des Adolf-Schmidt-Observatoriums für Erdmagnetismus in Niemegk, Akademie-Verlag Berlin, 1962

Schmidt, H.: Die Vermessung des Absoluthauses mit Protonenmagnetometern, Jahrbuch 1961 des Adolf-Schmidt-Observatoriums für Erdmagnetismus in Niemegk, Akademie-Verlag Berlin, 1963

Schmidt, H.: Zur Programmierung des Jahrbuch-Datenflusses für den Klein-Computer Cellatron SER 2, Jahrbuch 1965 des Adolf-Schmidt-Observatoriums für Erdmagnetismus in Niemegk, Akademie-Verlag Berlin, 1967

Wiese, H.: Die Anlage zur Registrierung der zeitlichen Gradienten des geomagnetischen Feldes in Niemegk, Erdmagnetisches Jahrbuch 1953, Akademie-Verlag Berlin, 1956

Wiese, H.: Ergebnisse der Beobachtungen am Adolf-Schmidt-Observatorium für Erdmagnetismus in Niemegk im Jahre 1954, Jahrbuch 1954 des Adolf-Schmidt-Observatoriums für Erdmagnetismus in Niemegk, Akademie-Verlag Berlin, 1957

Wiese, H.: Ergebnisse der Beobachtungen am Adolf-Schmidt-Observatorium für Erdmagnetismus in Niemegk im Jahre 1956, Jahrbuch 1956 des Adolf-Schmidt-Observatoriums für Erdmagnetismus in Niemegk, Akademie-Verlag Berlin, 1958

Wiese, H.: Ergebnisse der Beobachtungen am Adolf-Schmidt-Observatorium für Erdmagnetismus in Niemegk im Jahre 1957, Jahrbuch 1957 des Adolf-Schmidt-Observatoriums für Erdmagnetismus in Niemegk, Akademie-Verlag Berlin, 1960a

Wiese, H., Schmidt, H., Lucke, O., Frölich, F.: Geomagnetische Instrumente und Messmethoden. In Geomagnetismus und Aeronomie. Bd. 2 (Editor: Fanselau, G.), VEB Deutscher Verlag d. Wissenschaften, Berlin, 1960b

Wiese, H.: Ergebnisse der Beobachtungen am Adolf-Schmidt-Observatorium für Erdmagnetismus in Niemegk im Jahre 1959, Jahrbuch 1959 des Adolf-Schmidt-Observatoriums für Erdmagnetismus in Niemegk, Akademie-Verlag Berlin, 1962

Wiese, H.: Geomagnetische Tiefentellurik. Akademie Verlag Berlin, 1965

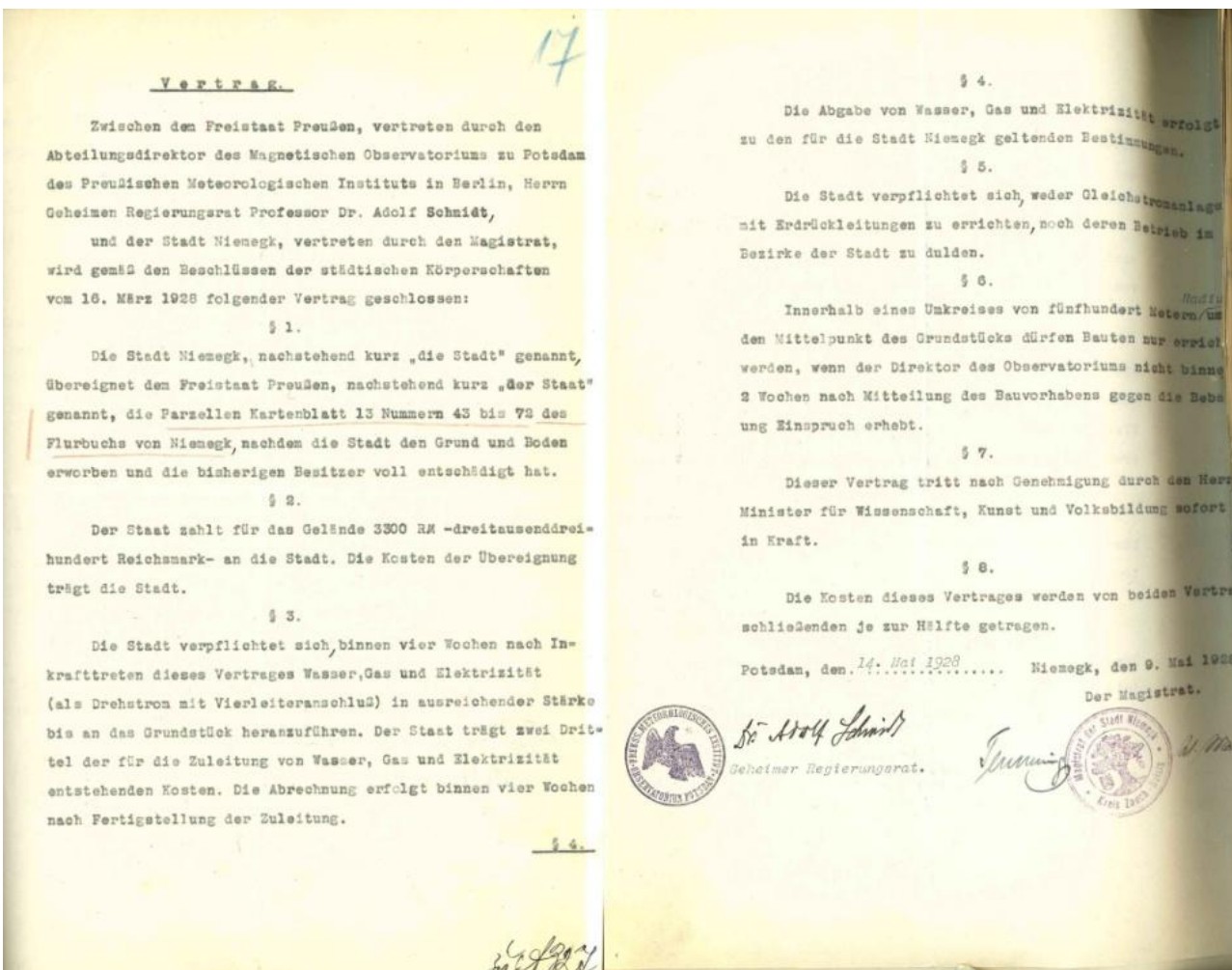

Fig. 1. Contract between the Free State of Prussia, represented by Adolf Schmidt and the magistrate of the town of Niemegk, represented by the mayor Paul Temming on the conditions for the undisturbed operation of the new observatory. Source: Helmholtz Centre Potsdam - GFZ

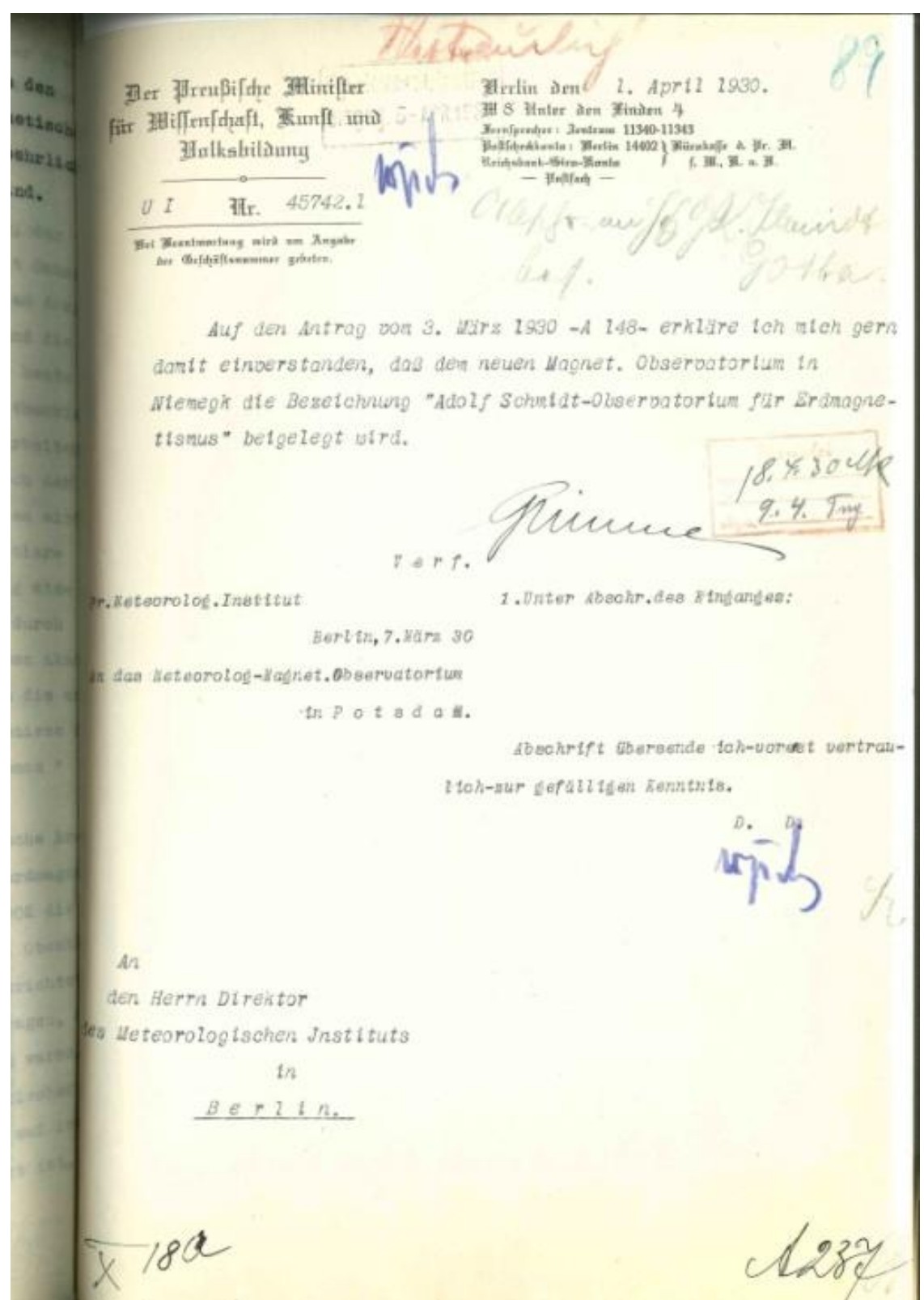

2  Fig. 2. Document from the Prussian Ministry for Science, Art and Education of 1 April 1928

3  attaching the new observatory the name "Adolf-Schmidt Observatorium für Erdmagnetismus

4  Niemegk" (Niemegk Adolf Schmidt Geomagnetic Observatory). Source: Helmholtz Centre

5  Potsdam – GFZ

2 Fig. 3. First page of the observatory guest book with Adolf Schmidt's inscription. Source:

3 Helmholtz Centre Potsdam – GFZ

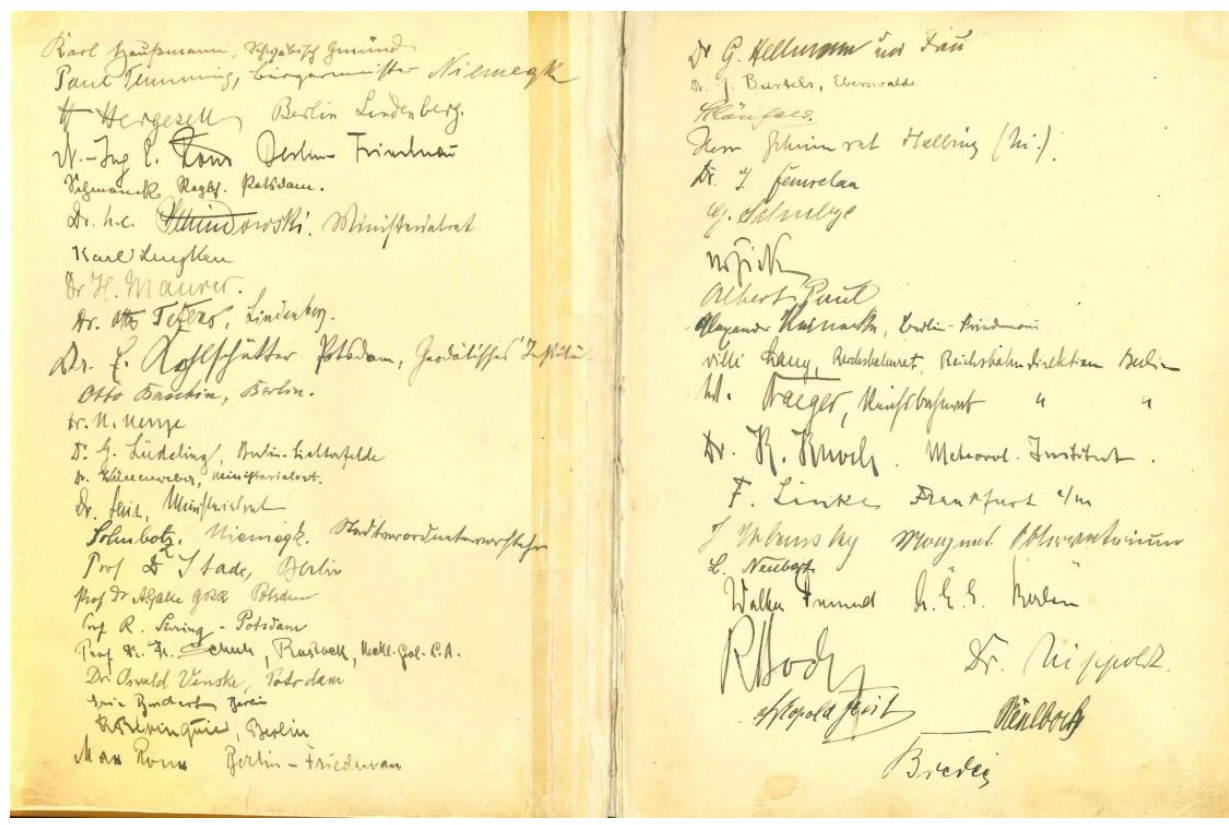

Fig. 4. Inscriptions of the participants of the observatory opening ceremony. Source: Helmholtz Centre Potsdam – GFZ

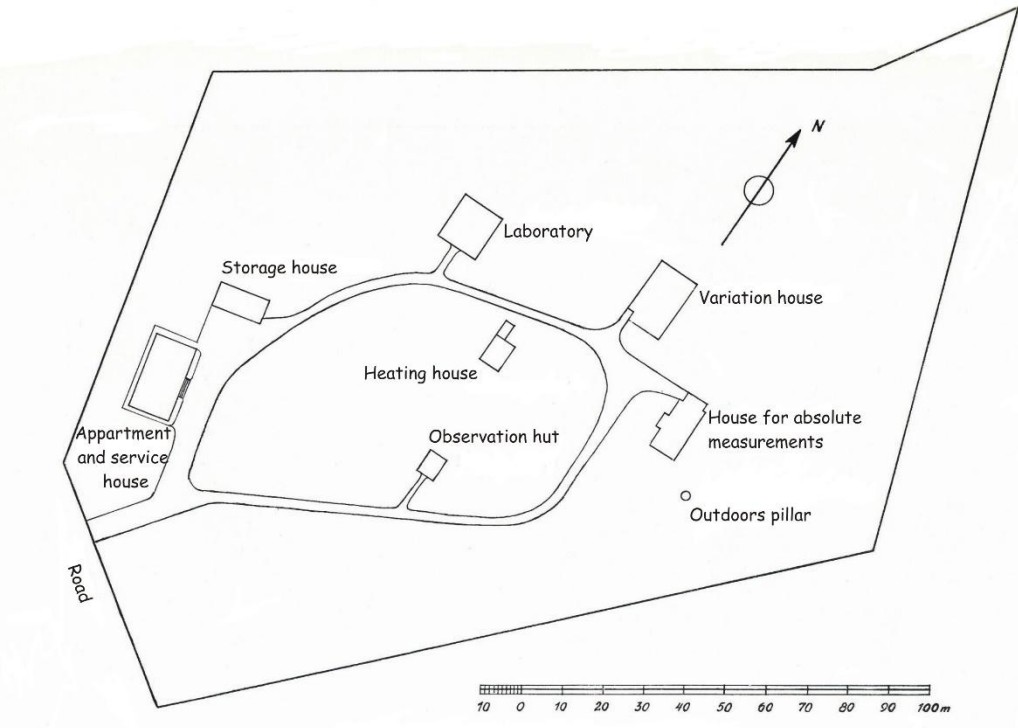

2 Fig. 5. Compound plan of the Niemegk Adolf Schmidt Observatory. Source: Bock, 1939

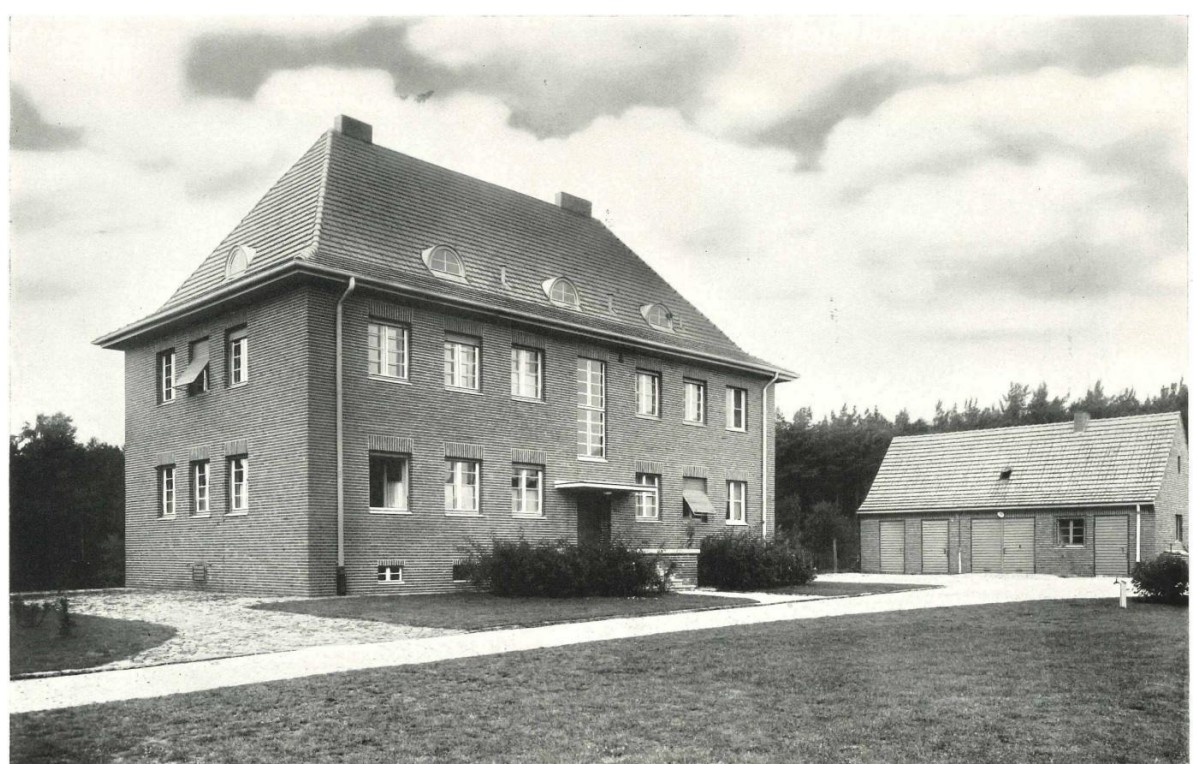

5 Fig. 6. Photo of the apartment and service house (left) and the storage house (right). Source:

6 Bock, 1939

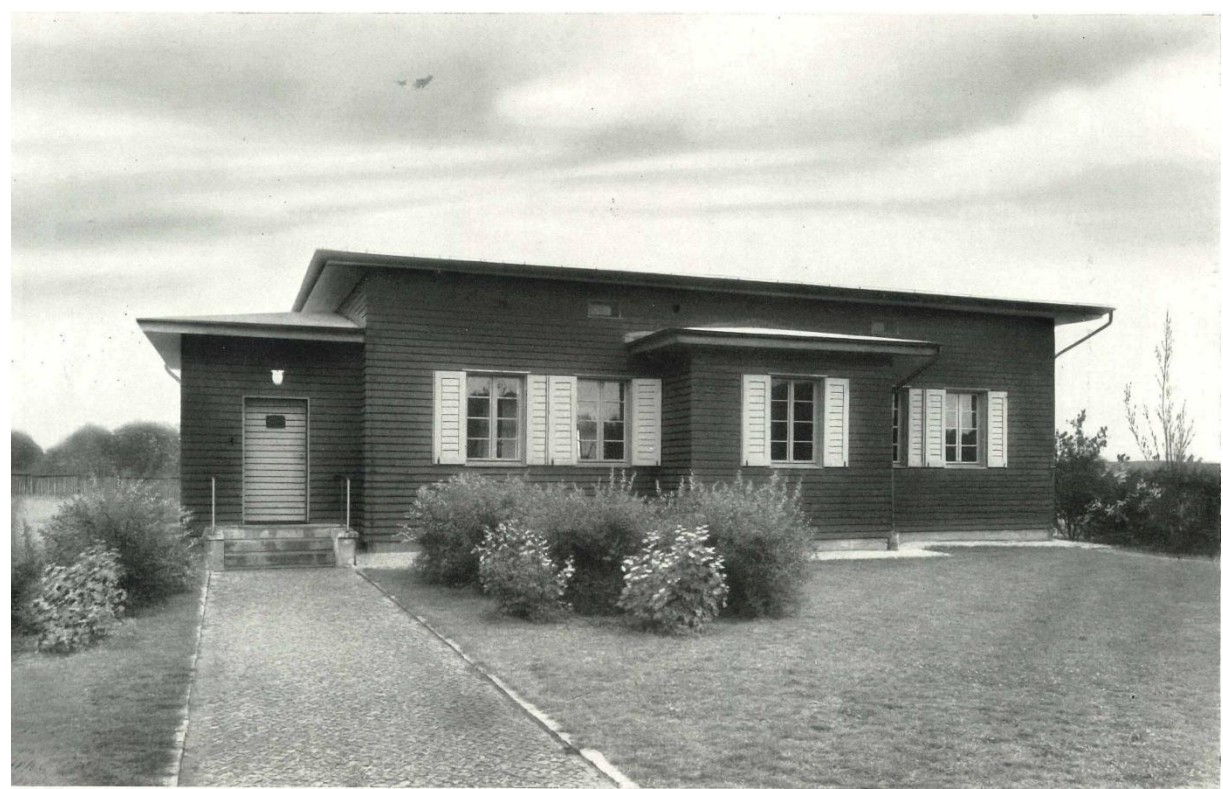

2    Fig. 7. Photo of the absolute house. Source: Bock, 1939

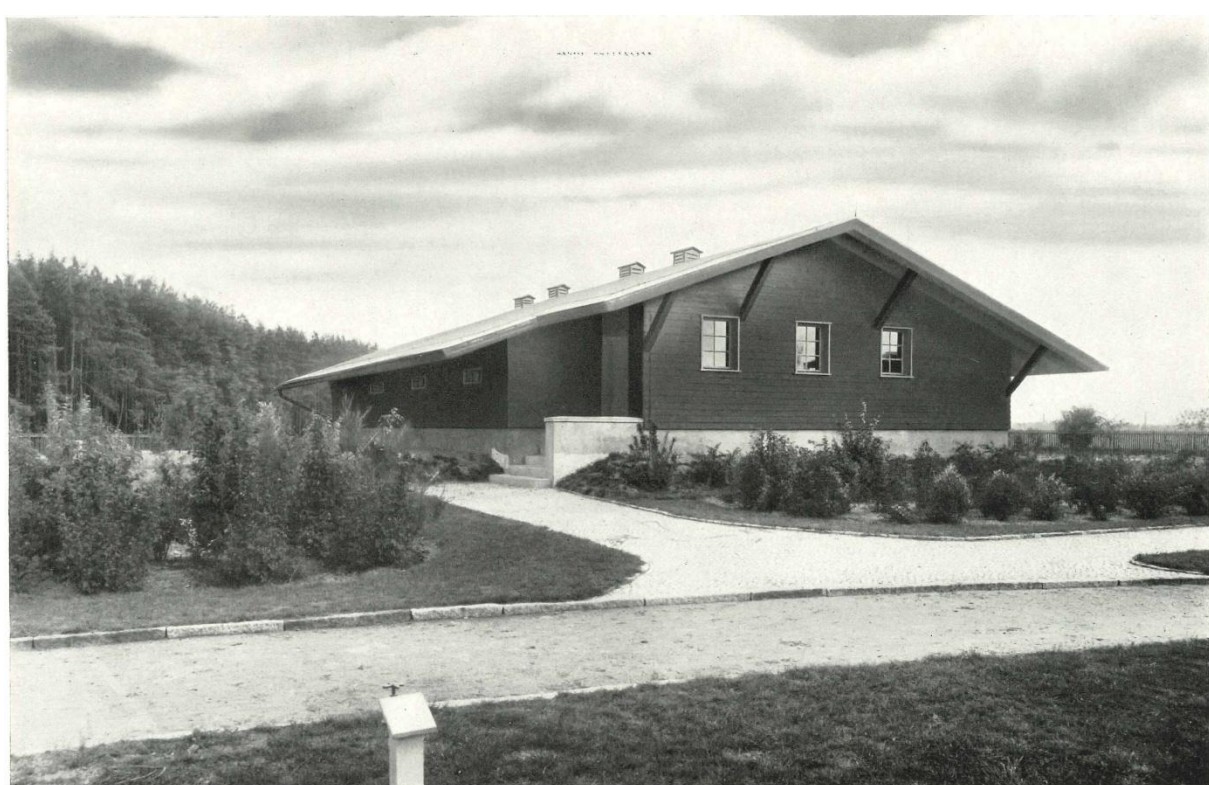

5    Fig. 8. Photo of the variation house. Source: Bock, 1939

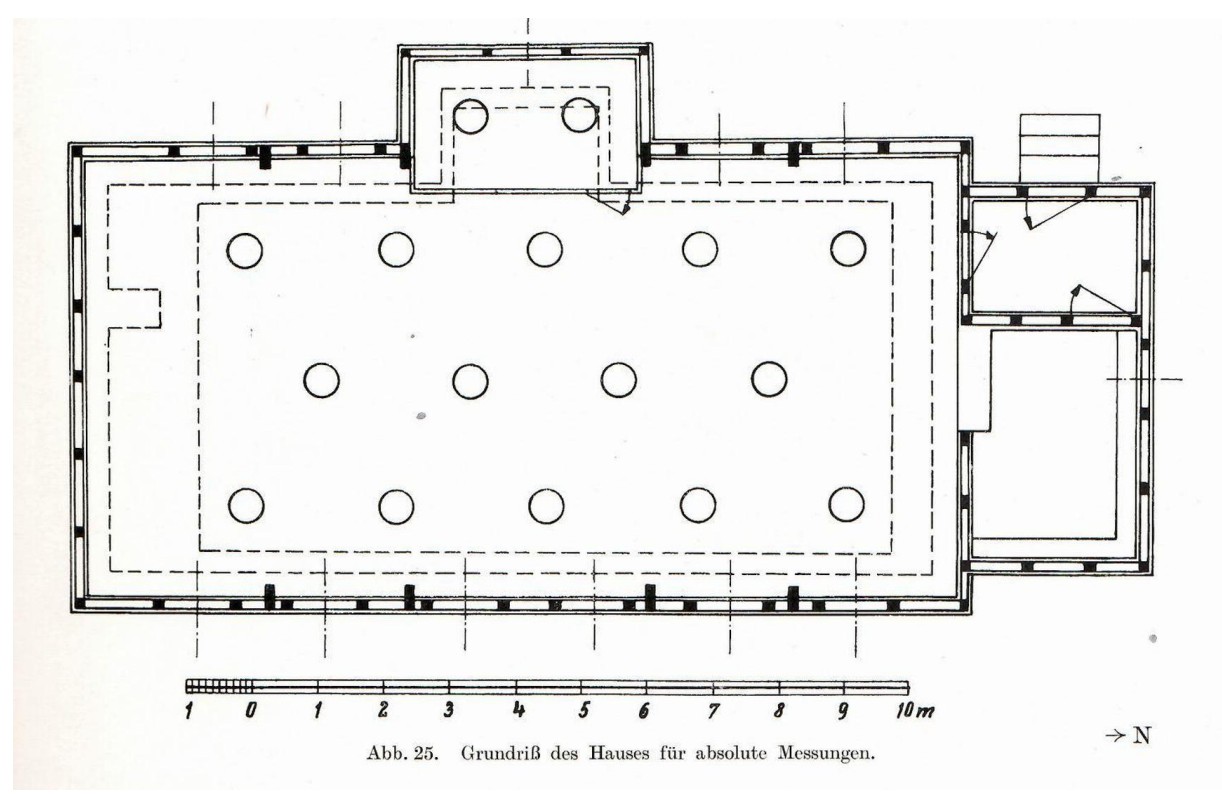

Abb. 25. Grundriß des Hauses für absolute Messungen.

2    Fig. 9. Ground plan of the absolute house. Source: Bock, 1939

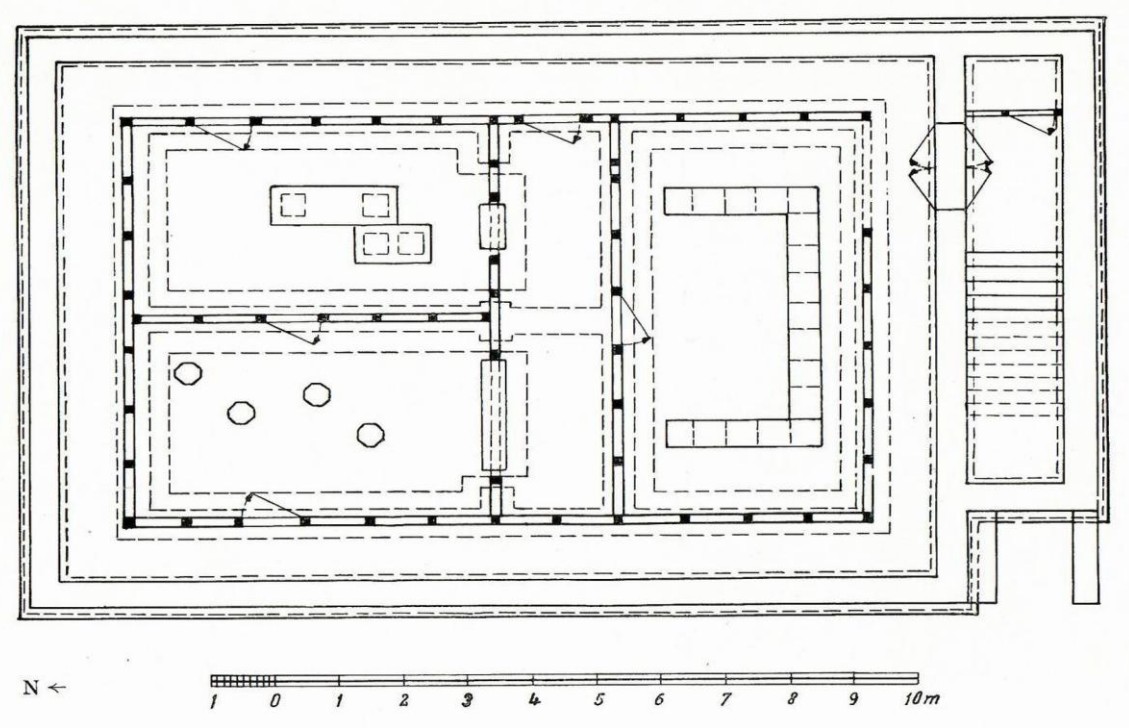

5    Fig. 10. Ground plan of the variation house. Source: Bock, 1939

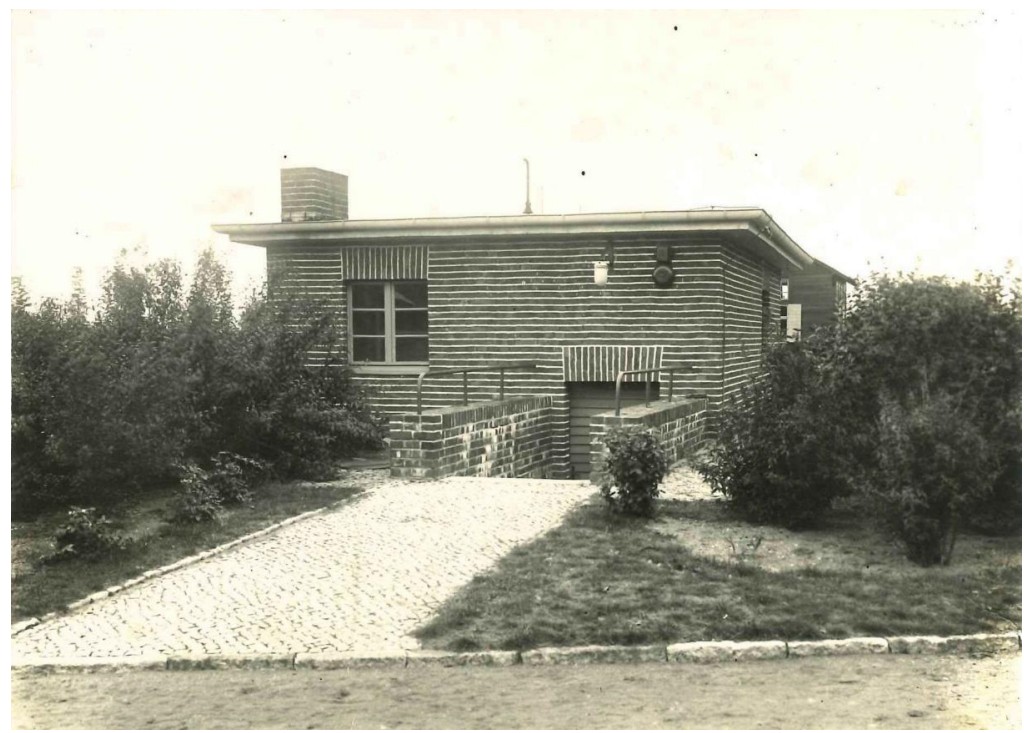

2    Fig. 11. Photo of the north-east corner of heating house. Source Helmholtz Centre Potsdam –

3    GFZ

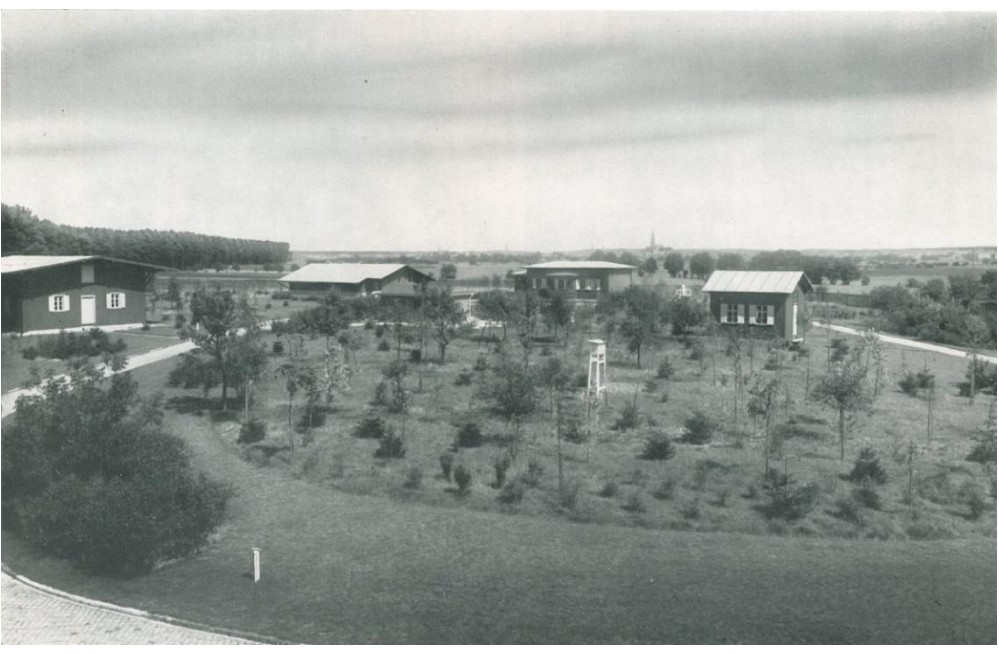

Fig. 12. Photo of the Niemegk Adolf Schmidt Geomagnetic Observatory compound, taken in

1933 from the apartment and service house. From left to right: laboratory (former Seddin

variation house), variation house, heating house (partly hidden by a tree), absolute house,

Niemegk church, outdoor pillar, observation hut (former Seddin absolute house). Source:

Bock, 1939

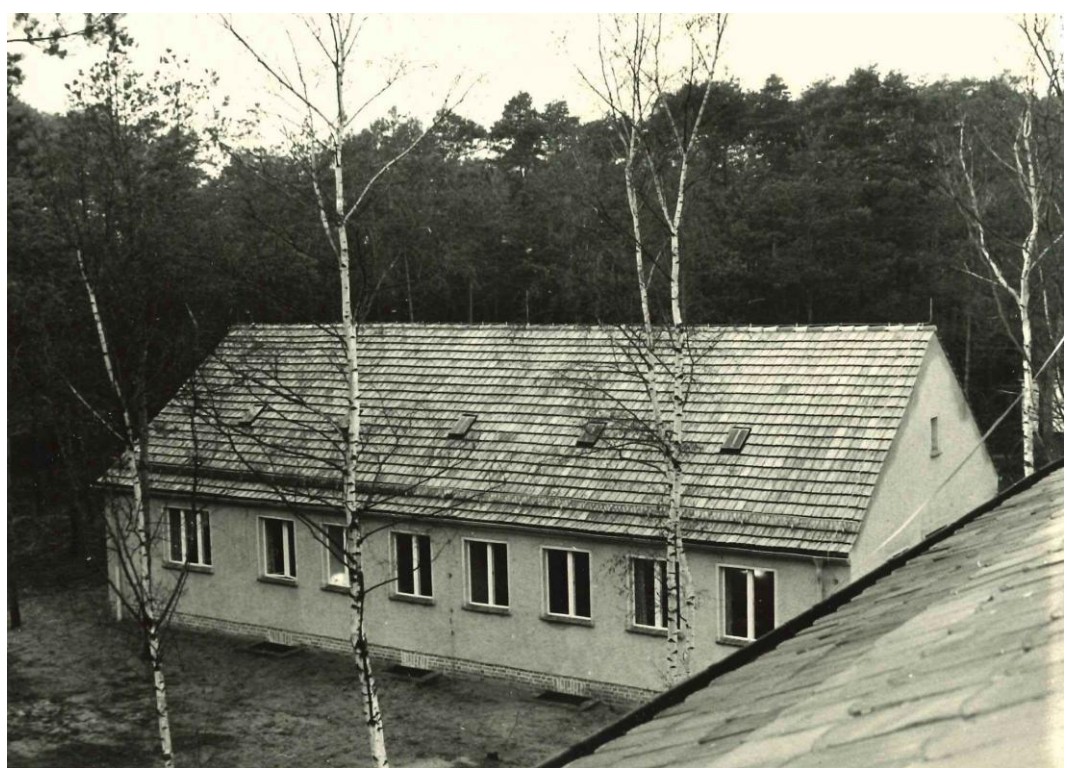

3    Fig. 13. View from the attic floor of the main building on the electric laboratory. Source:

4    Helmholtz Centre Potsdam – GFZ

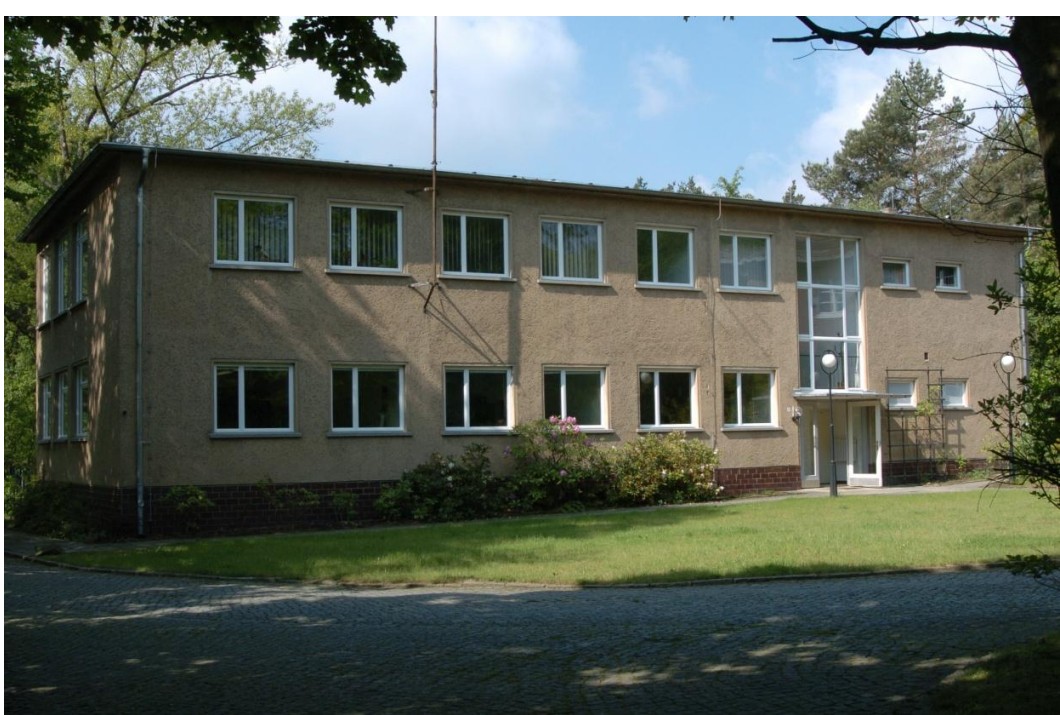

7    Fig. 14. Photo of the workshop building, view from the north-east. It was taken in 2005.

8    Source: Helmholtz Centre Potsdam – GFZ

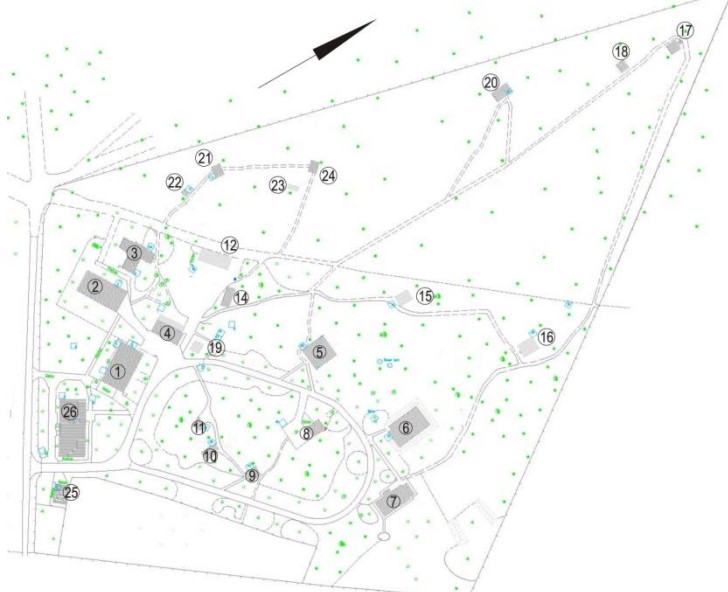

Fig. 15. Ground plan of the observatory compound, situation in 2003. Source: Helmholtz Centre Potsdam – GFZ

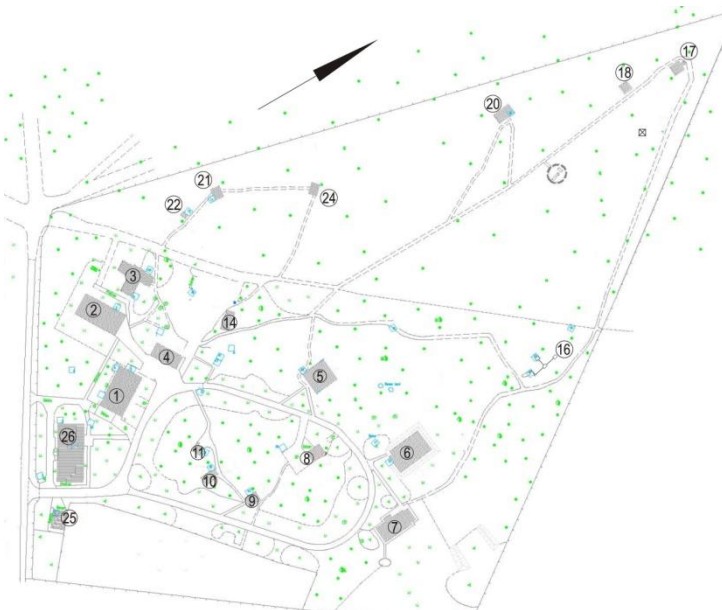

Fig. 16. Ground plan of the observatory compound, present situation. Source: Helmholtz Centre Potsdam – GFZ

| No. | Building |
|-----|----------|
| 1 | Main building |
| 2 | Electric laboratory |
| 3 | Computer centre |
| 4 | Storage house |
| 5 | Magnetic laboratory |
| 6 | Variation house |
| 7 | Absolute house |
| 8 | Heating house |
| 9 | Small hut |
| 10 | Adjustment hut |
| 11 | Thermal adjustment hut |
| 12 | Garage |
| 14 | Equipment shed |
| 15 | Proton magnetometer hut |
| 16 | Control hut No. 1 |
| 17 | Coil hut No. 1 |
| 18 | Control hut No. 2 |
| 19 | Measurement centre |
| 20 | Teluric hut |
| 21 | Coil hut No. 2 |
| 22 | Small control hut |
| 23 | Control hut No.3 |
| 24 | Coil hut No.3 |
| 25 | Power unit house |
| 26 | Workshop building |

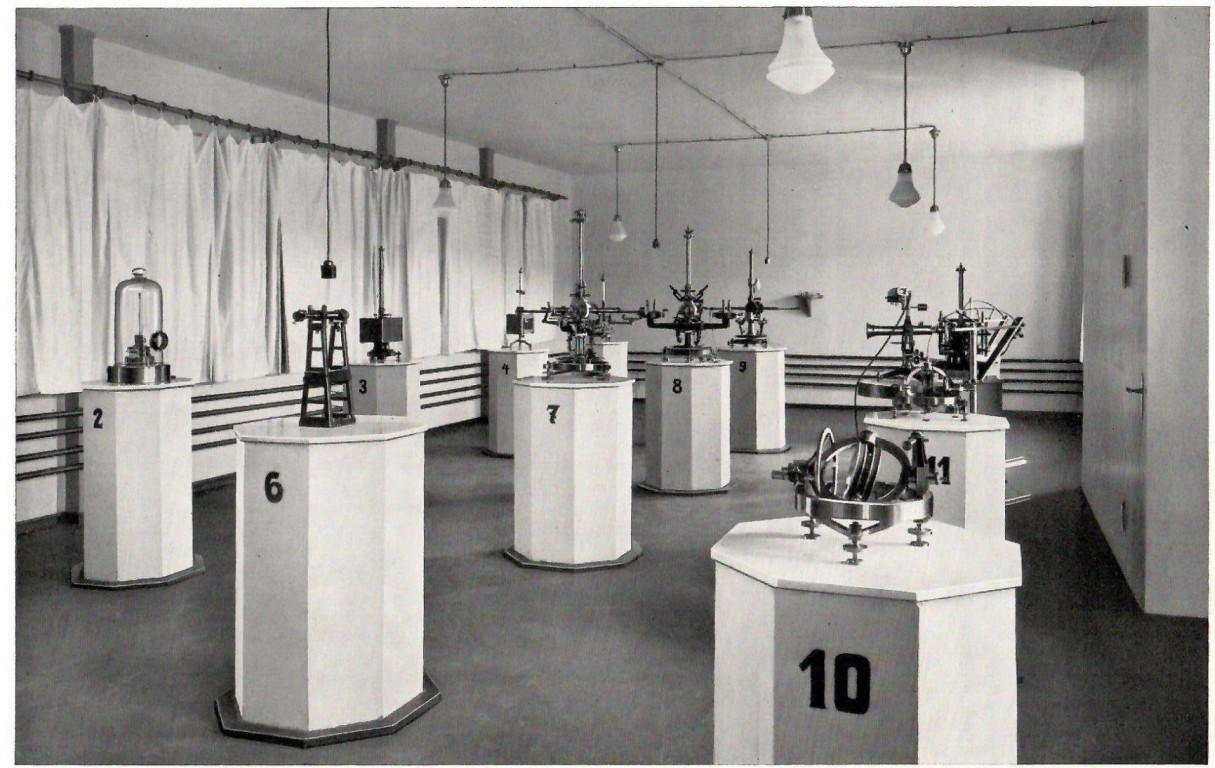

| Pillar No. | Instrument |
|---|---|
| 2 | Galvanometer for the earth inductors |
| 3 | Oscillation box Wanschaff |
| 4 | Oscillation box Schulze (Fürstenfeldbruck) |
| 5 | Theodolit Schulze No. 65 (Fürstenfeldbruck) |
| 6 | Collimator (azimuth mark in case of invisible towers) |
| 7 | Theodolit Bamberg |
| 8 | Theodolit Schmidt |
| 9 | Theodolit Wanschaff |
| 10 | Earth inductor Schulze No. 550 (Fürstenfeldbruck) |
| 11 | Earth inductor Schulze No. 1 |
| 13 | Earth inductor Schulze No. 65 |
| 14 | Journey theodolit Schulze No. 541 |

Fig. 17. Interior view of the absolute house in 1932. The table contains the assignment of the visible instruments to the pillars. Source: Bock, 1939

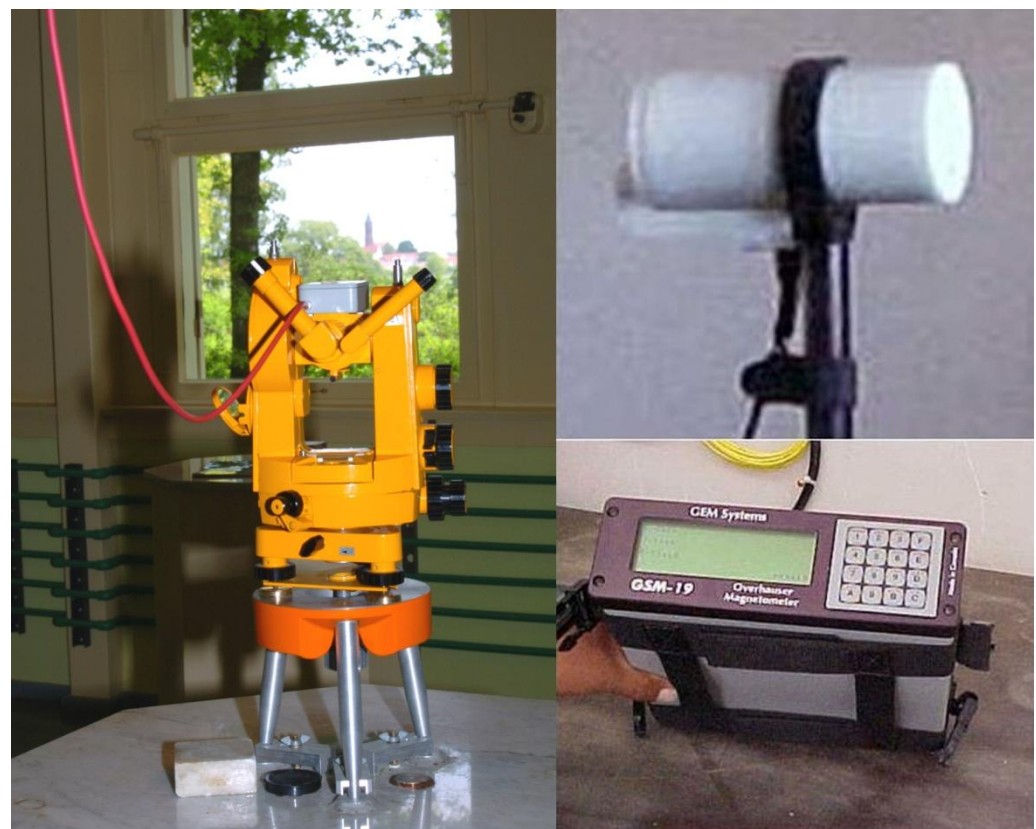

2  Fig. 18. DI-flux on pillar No. 8 of the absolute house with the Niemegk church tower in the

3  background (left) and Overhauser proton magnetometer GSM19 (right, sensor up and

4  electronic unit down). Source: Helmholtz Centre Potsdam – GFZ.

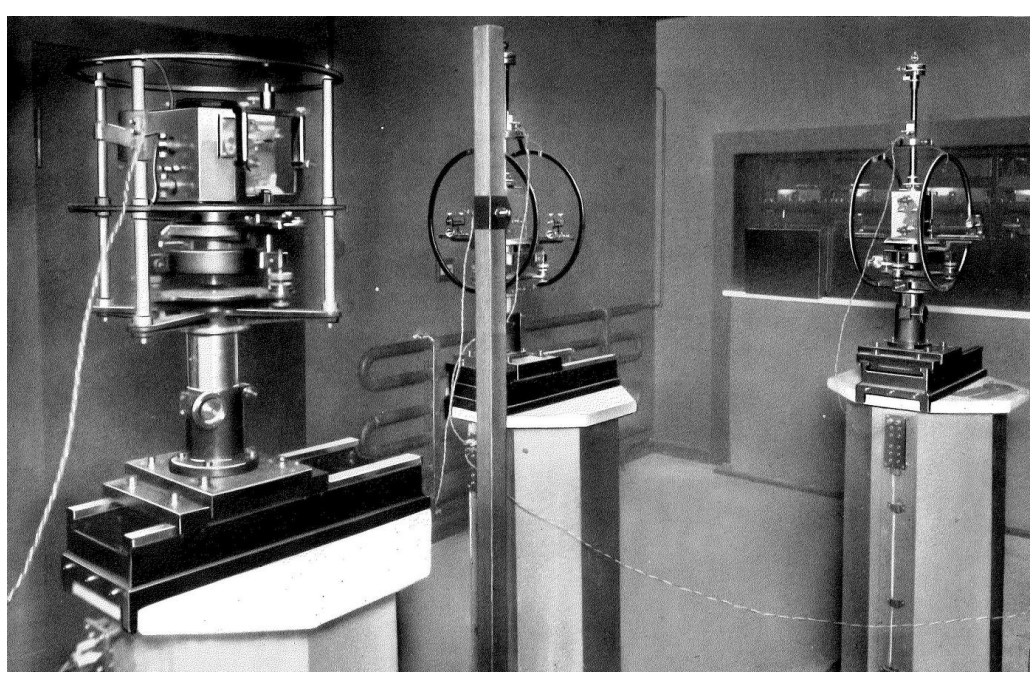

7  Fig. 19. Photo of the interior of the north-west room of the variation house. Source: Bock,
8  1939

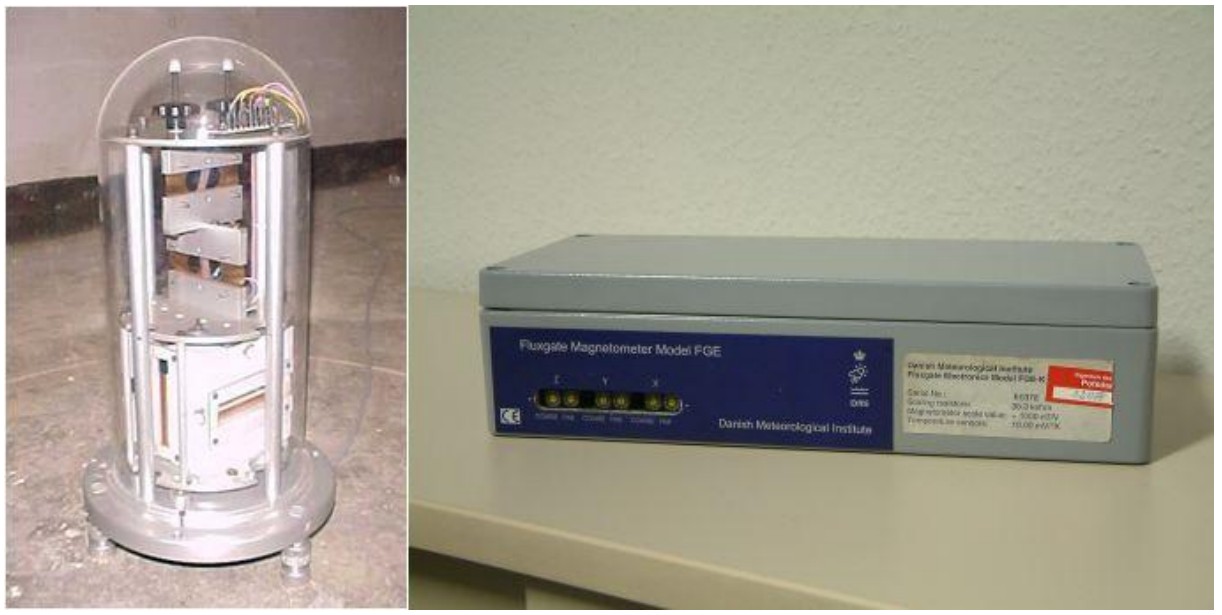

Fig. 20. Fluxgate magnetometer FGE sensor (left) and electronic unit (right). Source: Helmholtz Centre Potsdam – GFZ.

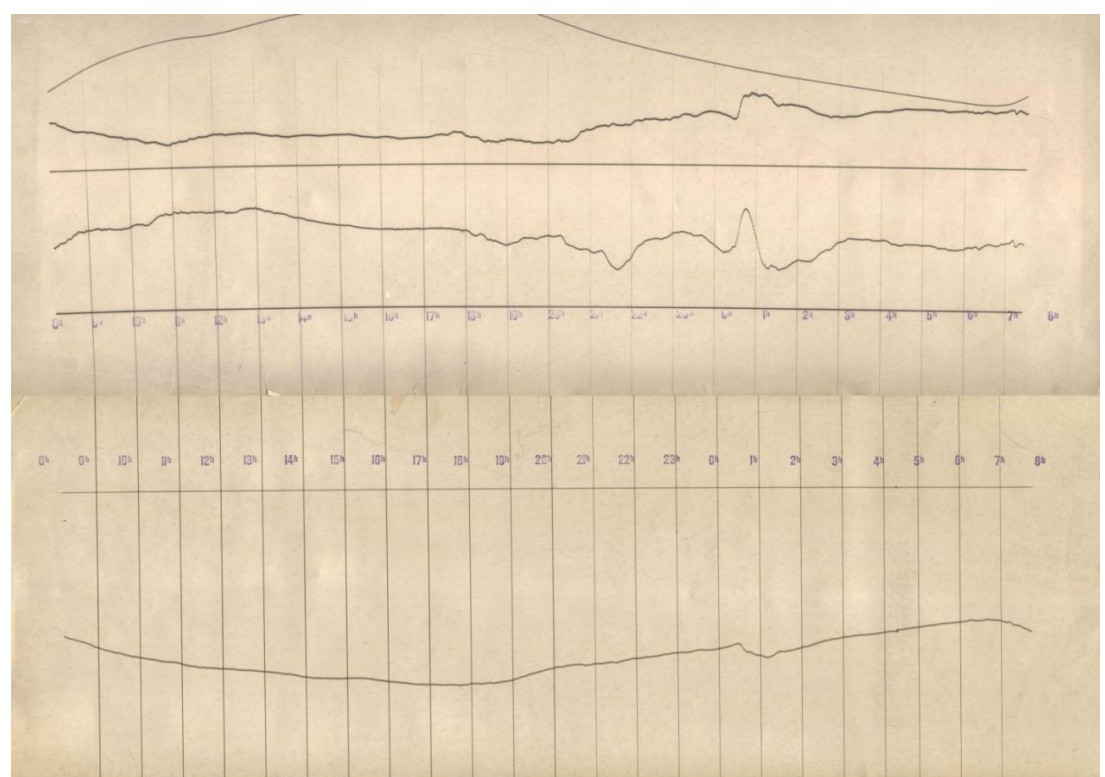

Fig. 21. One of the first photographic recordings of the horizontal intensity and declination (top) and the vertical intensity (bottom) of the time interval 25 March 1931 at 08:00 till 26 March 1931 at 07:20 (Greenwich local mean time)  taken at the Niemegk Adolf Schmidt Geomagnetic Observatory. Source: Helmholtz Centre Potsdam – GFZ.

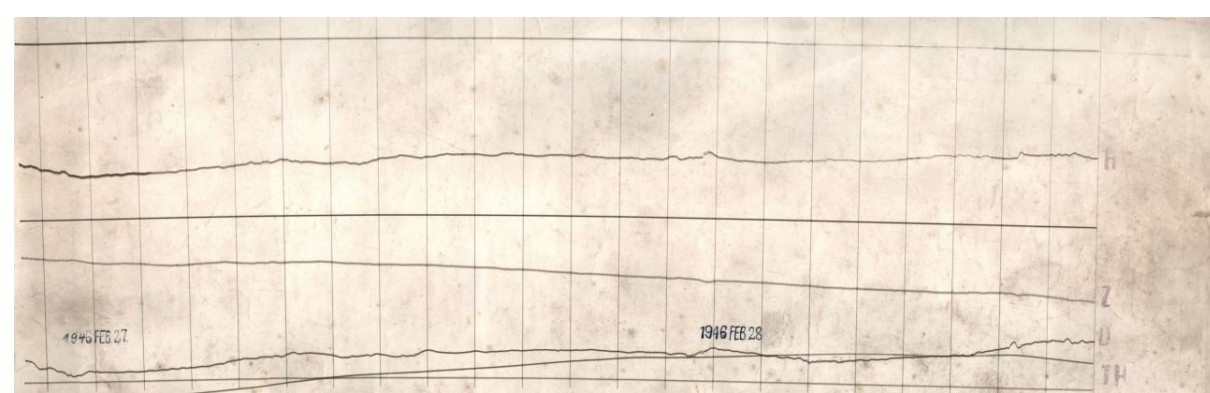

Fig. 22. First photographic recordings after the operation gap caused by World War II of the

horizontal (H) and vertical (Z) intensity and declination (D) of the time interval 27 February

1946 at 10:30 till 28 February 1946 at 9:00 (Greenwich local mean time)  taken at the

Niemegk Adolf Schmidt Geomagnetic Observatory. Source: Helmholtz Centre Potsdam –

GFZ.

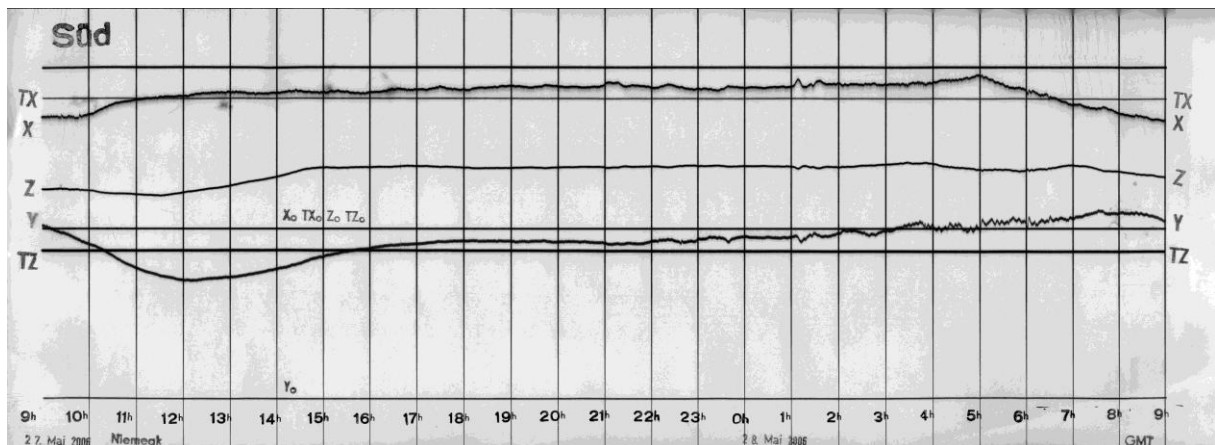

10 Fig. 23. Last photographic recordings of the north (X), east (Y) component and the vertical

11 (Z) intensity of the time interval 27 May 2006 at 09:00 till 28 May 2006 at 9:00 (Greenwich

12 local mean time)  taken at the Niemegk Adolf Schmidt Geomagnetic Observatory. Source:

13 Helmholtz Centre Potsdam – GFZ.

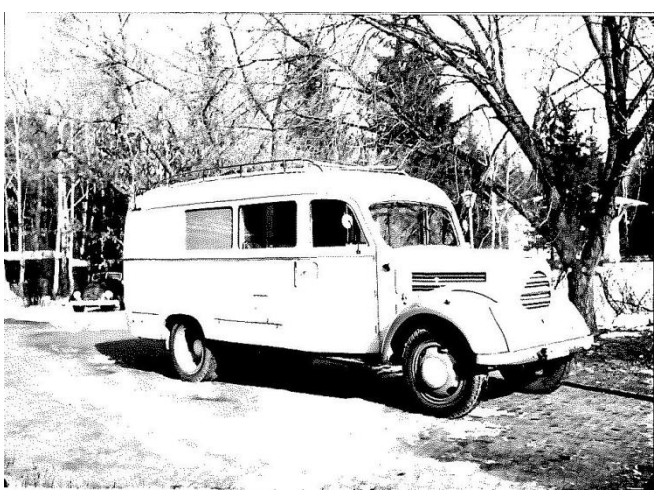

2 Fig. 24. Photo of the survey van Phänomen Granit 30K. Source: Helmholtz Centre Potsdam –

3 GFZ.

6 Fig. 25. Heinrich Soffel, the National Representative of Germany for IAGA (right), hands

7 over the Long Service Award of IAGA to Walter Zander (left). Source: IAGA News No. 32,

8 https://iaga-aiga.org/data/uploads/pdf/newsletter/iaganews_32_1993.pdf

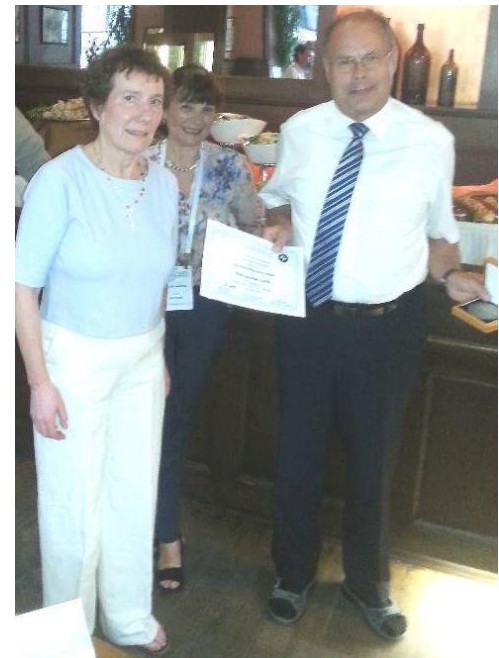

2 Fig. 26. From left to right: Kathy Whaler (IAGA President 2011-2015), Mioara Mandea

3 (IAGA Secretary General 2009-2019) and Hans-Joachim Linthe after receiving the IAGA

4 Long Service Medal. Source: Mandea, 2015,

5 https://iaga-aiga.org/data/uploads/pdf/newsletter/iaganews_52.pdf

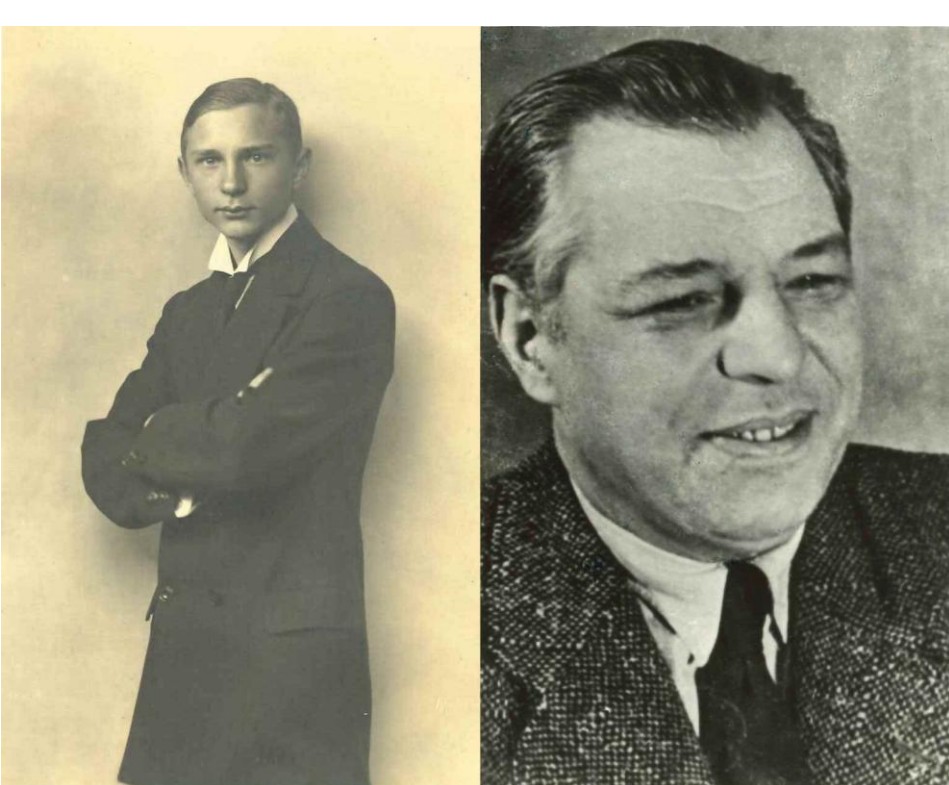

8 Fig. 27. Julius Bartels' portrait (left) and Richard Bock's portrait (right). Source: Helmholtz

9 Centre Potsdam – GFZ

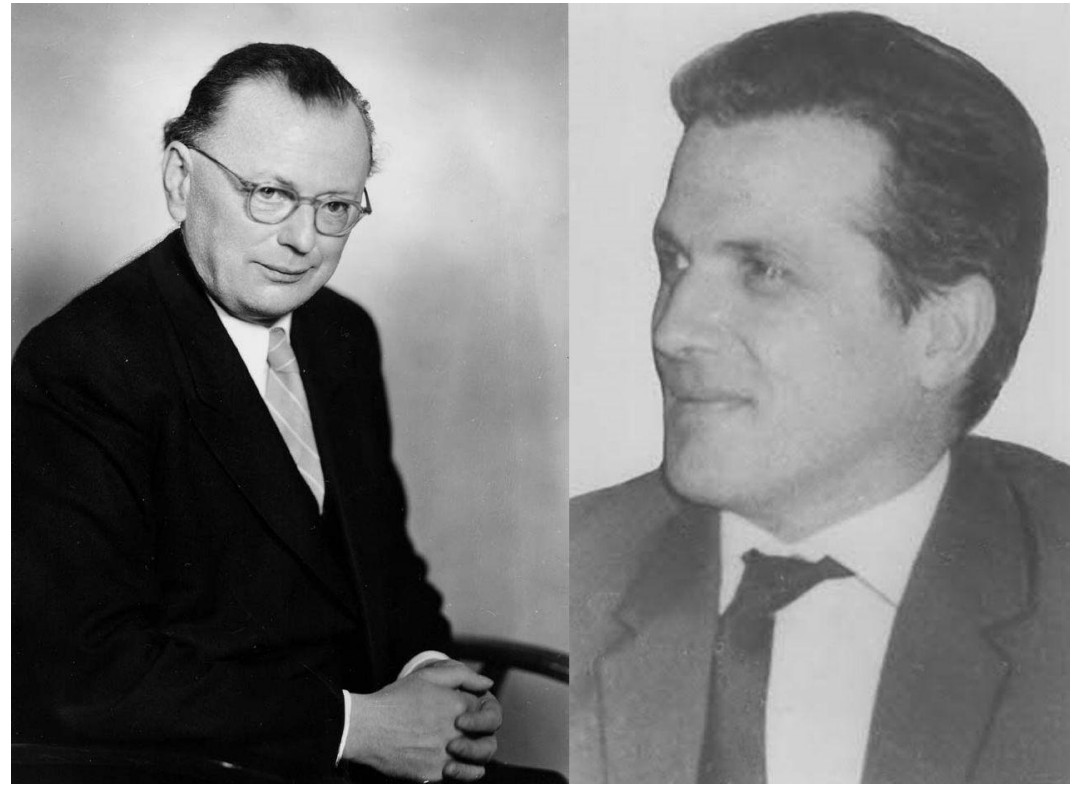

3 Fig. 28. Gerhard Fanselau's portrait (left) and Horst Wiese's portrait (right). Source:

4 Helmholtz Centre Potsdam – GFZ

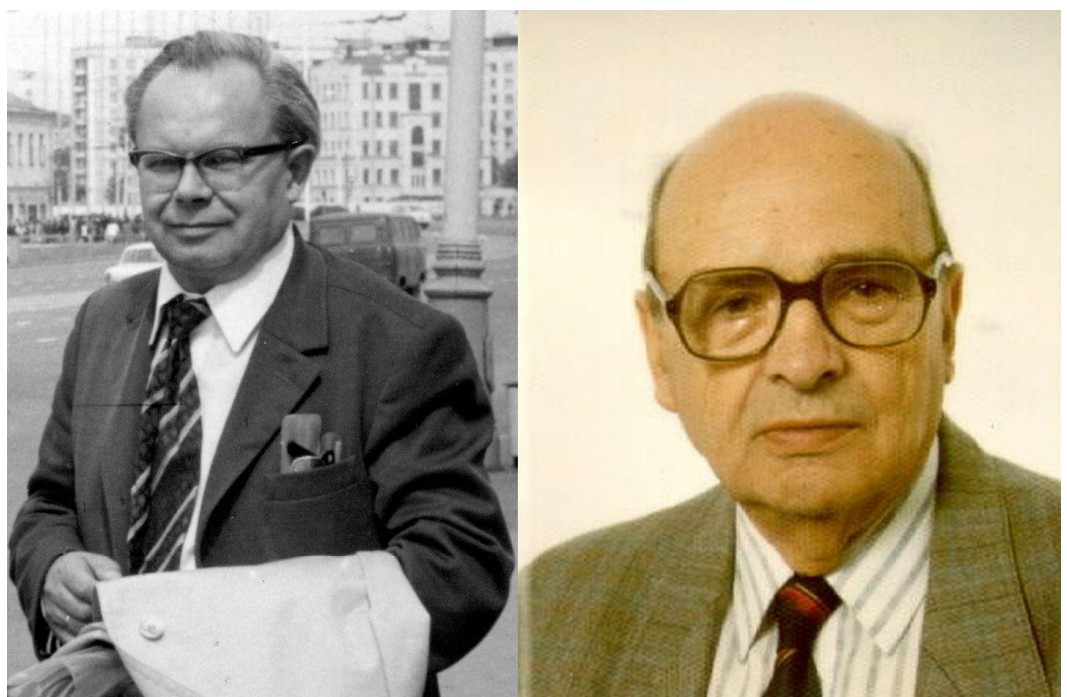

7 Fig. 29. Herbert Schmidt's portrait (left) and Klaus Lengning's portrait (right). Source:

8 Helmholtz Centre Potsdam – GFZ

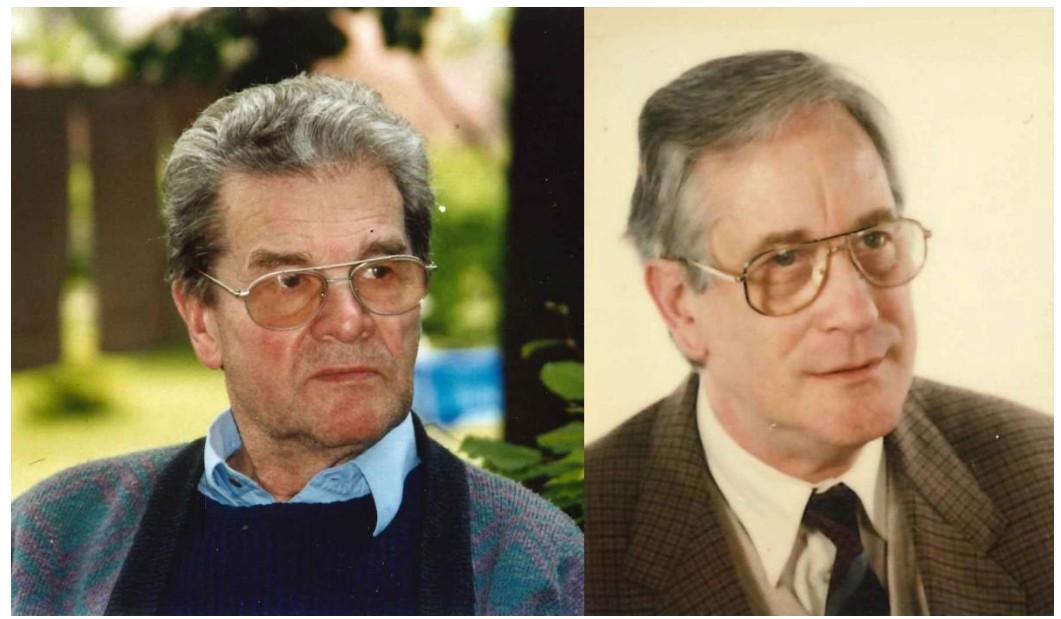

2    Fig. 30. Armin Grafe's portrait (left) and Adolf Best's portrait (right). Source: Helmholtz

3    Centre Potsdam – GFZ

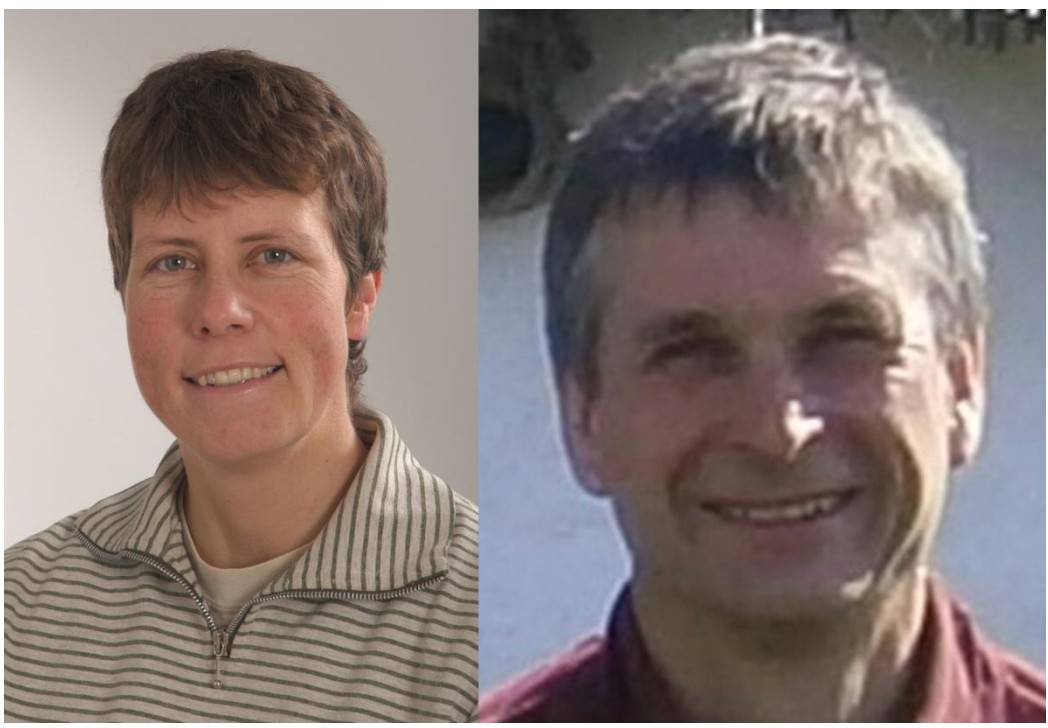

6    Fig. 31. Monika Korte's portrait (left) and Richard Holme's portrait (right). Source:

7    Helmholtz Centre Potsdam – GFZ

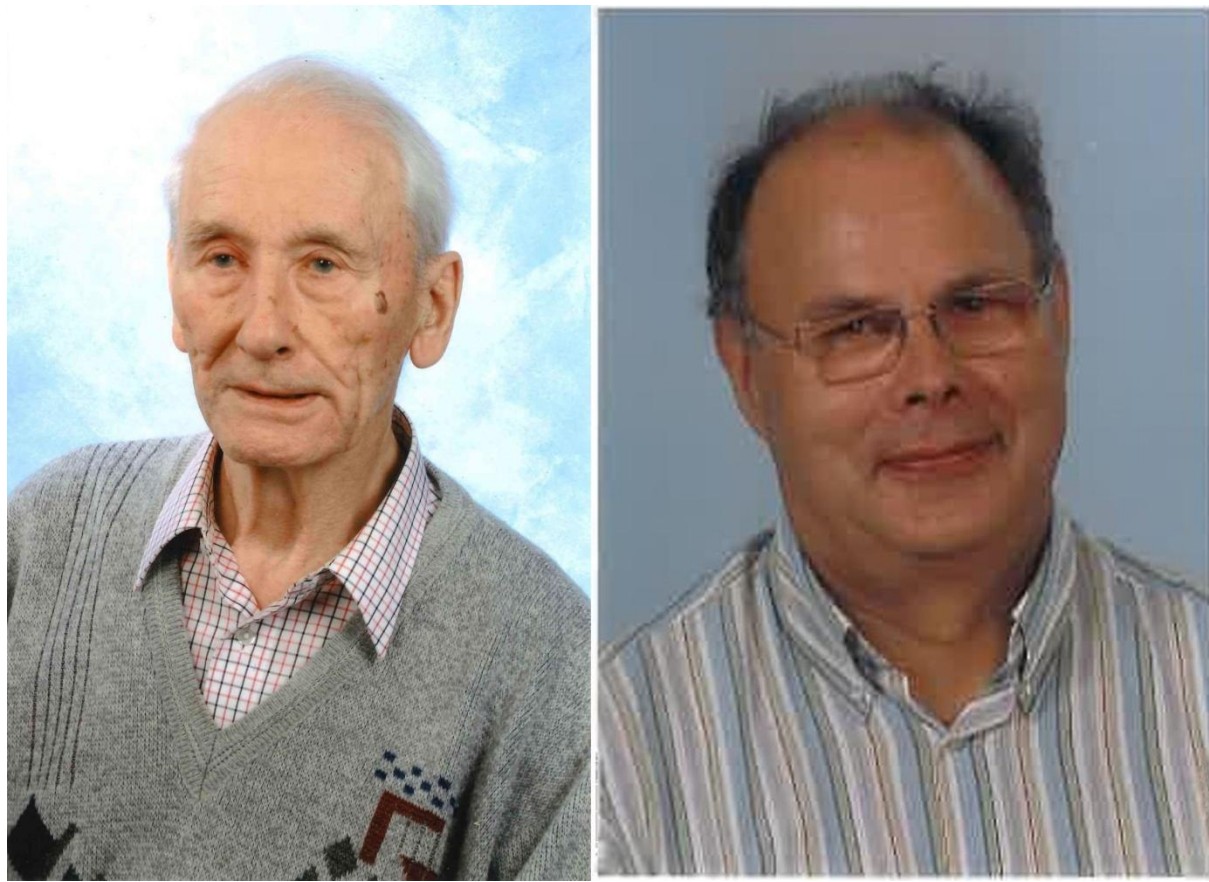

2  Fig. 32. Eberhard Ritter's portrait (left) and Hans-Joachim Linthe's portrait (right). Source:

3  Helmholtz Centre Potsdam – GFZ

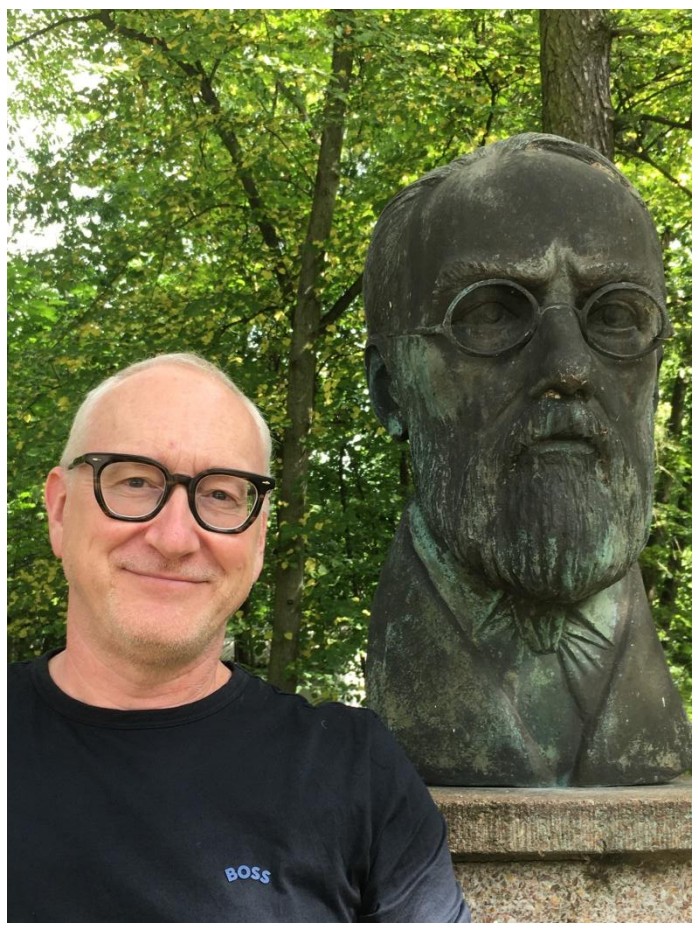

2 Fig. 33. Jürgen Matzka's portrait next to the Adolf Schmidt bust at the Niemegk observatory

3 compound. Source: Helmholtz Centre Potsdam – GFZ

