# Peer review of "History of the Potsdam, Seddin and Niemegk Geomagnetic"

_History of Geo- and Space Sciences, 2024_

## Author Comment (AC1)

Dear referee,

Thank you very much for your patience while commenting my paper. Your comprehensive comments are very helpful for me to improve the paper. I went through your comments one by one. Here ae my responses:

I agree that the paper is rather long. But it is difficult to make it shorter. Please consider that the observatory exists since almost 100 years and worked over a lot of topics, exceeding the poor ordinary tasks of a geomagnetic observatory, I do not want to delete details, which ae important to a number of readers. To make the reading easier, I structured the text as good as possible and put some information into tables. But I will try to follow your suggestions of putting details into appendixes.

I am not a native English speaker. Therefore, a language improvement is necessary. I expect, Copernicus will ask an expert for this task – as my experience shows it from the edition process of parts 1 and 2.

"The very well-known and outstanding scientists for geomagnetism, Max Eschenhagen, Adolf Schmidt, Julius Bartels, Gerhard Fanselau and Horst Wiese directed among others, in historical sequence, the three observatories." I took over this phrase completely, thank you for it.

"...and is currently still in operation, acting as the German reference observatory for geomagnetism." I only took over the first part of the sentence. The second part is rather arrogant with respect to the merits of the Fürstenfeldbruck observatory.

I followed your advise to extend the introduction. You may read the new version in the upgraded paper.

It would be very much time consuming to include comparison graphics of Seddin and Niemegk. The photographic recordings are available, but the Niemegk ones of 1931 are fragmentary, the same is the case for Seddin beginning with 1932. Further more the not published data are not available in digital form. The procedure of processing is very time consuming. I am afraid to miss the deadline of submitting the revised paper, therefore, I decided to omit your suggestion in this regard. I hope on your understanding.

I do not agree to shift parts of chapters 2.2.1 and 2.2.2. Each spans over 3 or 4 pages and to detach parts destroys the context. But I will follow your suggestion including functional descriptions of the instruments in an appendix.

I further exceeded parts of chapter 2.3 (new notation), which do not belong to the ordinary observatory duties and put them into an appendix.

I enlarged Fig. 1-4 hoping to improve the readability adequately.

The difference of Fig. 15 and 16 is the number of buildings. At page 6, 16 can be found the explanation:

"Further buildings were constructed at different times for different purposes. Finally 26 buildings existed on the observatory compound. Fig. 15 shows the ground plan of 2003. One of the 26 buildings did not any more exist at this time. Three more buildings, which were not any more in use, were removed in 2004. Fig. 16 shows the present ground plan of the observatory compound."

Fig. 24: In the text is written:

A van "Phänomen Granit 30 K" (Fig. 24 shows a photo of it) was in use for all expeditions. The van was completely equipped with any necessary instruments.

The photo was taken at the observatory compound. I did not consider this to be important to be mentioned.

Do you consider to extend this?

I followed all your further comments. I did not each modification in detail, because the revision is rather comprehensive, following your advises, Especially the shifting of a number of details from the normal text into appendixes changed the format of the paper extremely, but for my opinion it improved the paper. Thank you very much.

---

## Author Comment (AC2)

Dear referee,

Thank you very much for your patience while commenting my paper. Your comprehensive comments are very helpful for me to improve the paper. I went through your comments one by one. Here ae my responses:

I considered social and scientific aspects already from the beginning, for instance the construction problem of the variation house, the situation at the end of World War II and the problems of the German reunification on the personal. May be this is not sufficient in your opinion, but I do not know what is missing.

I am not a native English speaker. Therefore, a language improvement is necessary. I expect, Copernicus will ask an expert for this task – as my experience shows it from the edition process of parts 1 and 2.

Following the comments of referee1 I reformatted the paper completely. I very hope, that you can agree in the new structure.

I extended the information regarding the take over of the Kp index service following your comment.

Thank you again for taking your time to act as referee.

---

## Author Response (AR2)

Dear referee 1,

Thank you very much for your extremely helpful comments on my paper. I followed your suggestions as follows. Please find my answers in green:

**Suggestions for revision or reasons for rejection**

(visible to the public if the article is accepted and published)

Review comments on the revised manuscript 10.5194/hgss-2024-2. "History of the Potsdam, Seddin and Niemegk Geomagnetic Observatories - Part 3: Niemegk"
H.-J. Linthe

After the revision the manuscript has improved significantly. Now the readability of the paper has increased a lot. More technical and administrative parts are moved to Appendices. After some minor adjustments, see details below, the article should be ready for publication in HGSS.

Pg. 2, line 21, For completeness, it should be added: (now termed: GFZ - German Research Centre for Geosciences)
I added the present correct name of the GFZ.

After line 26, It is good practice in papers to present at the end of the Introduction a short overview of the sections to follow in the article. That should also appear here. In particular an announcement and justification of the Appendices should be added and a listing of their titles.
I added the titles of the chapters and appendices.

Pg. 7, after line 4, A note should be added here to the short description in Annex I of the absolute measurement approach in those days.
Similarly after line 33, the note to Appendix I about the procedure of modern absolute measurements.
I did as you suggest at the following positions:
-   Pg. 7 line 15/16 (was already present in the previous version of the manuscript)
-   Pg. 8 line 25/26
-   Pg. 11 line 26/27

Pg. 8, line7, Make clear, for witch purpose were the lamps needed, for illuminating the room or for the photographic registration?
I did at Pg. 9 line 1-3

Line 3: Mention the purpose of the quick-run recordings. Probably, it was the interest in magnetic pulsation measurements.
You did not state the page of this. I could not find "quick-run recordings" at pages 8 and 9. Page 9 line 1-3 describes the storm variometer. Its purpose was the visibility of magnetic storms by using a reduced scale value. The recording tracks of "normal" magnetograms exceed the photographic paper and information is lost.

Pg. 9, lines 25ff: What were the special features of these three-component recordings?

Without this knowledge the sentence contains no information for the reader.
I added "paper-economizing" at Pg. 10, line 20 to explain the purpose.

Pg. 13, line2, The meaning of the sentence " In 1965 a survey of the total intensity..." is not clear, what was the purpose of the survey, what the outcome?
I hope the wording "In 1962 a survey of the total intensity in the absolute house was carried out (Schmidt, 1963) to find out magnetic anomalies. Neglectable anomalies were found." will explain it sufficiently. I further added the sentence "The new survey confirmed the results of (Schmidt, 1963)." at Pg. 15 line 6/7.

Pg. 17, line 7ff, The description of fluxgate should be improved, e.g.: "The working principle of this instrument is based on the saturation of a transformer core. In this situation primarily odd harmonics of the excitation frequency are produced. When in addition to the alternating excitation field also a DC components of the environmental magnetic field acts on the core, also even harmonic signals appear. By means of suitable electronic circuits these even harmonics are detected and converted into an output voltage proportional to the Earth's magnetic field component in the direction of the sensor core."

Thank you very much for this much better description. I inherited your wording completely.

Lines 11ff, Also for the proton magnetometer it could be improved, e.g.: " It consists of a vessel filled with a proton-rich liquid, a surrounding coil, and electronic control circuits. The coil is used in two modes: In the first one a strong DC current flows through the coil. The induced magnetic field forces the protons in the liquid sample to align their spin axis along the coil field. After switch off of the DC current the protons start to reorient their spin axis towards the ambient magnetic field by performing a precession motion. The resulting precession frequency is proportional to the ambient magnetic field strength. In this phase the coil is used to pick up the proton presession signal and the electronics measures the derived frequency. Best results are achieved when the direction of the applied DC field is approximately perpendicular to the Earth's magnetic field."

Thank you very much for this much better description. I inherited your wording completely.

Lines 21ff, For the optically pumped magnetometers at least some references should be added.

I added 2 references at Pg. 19 line 3/4.

**For final publication, the manuscript should be**
**accepted as is**.
accepted subject to **technical corrections**.

accepted subject to **minor revisions**.

reconsidered after **major revisions**:

**rejected**.

Were a revised manuscript to be sent for another round of reviews:

I would be willing to review the revised manuscript.

I would not be willing to review the revised manuscript.

**Suggestions for revision or reasons for rejection**

(visible to the public if the article is accepted and published)

The revised version reads much better. There is fair scope for English language editing.

Dear referee 2,

Thank you very much for the renewed review of my manuscript. I highly appreciate your judgment on my revised manuscript.